# Feeding Twins with Human Milk and Factors Associated with Its Duration: A Qualitative and Quantitative Study in Southern Italy

**DOI:** 10.3390/nu13093099

**Published:** 2021-09-03

**Authors:** Pasqua Anna Quitadamo, Laura Comegna, Giuseppina Palumbo, Massimiliano Copetti, Paola Lurdo, Federica Zambianco, Maria Assunta Gentile, Antonio Villani

**Affiliations:** 1Neonatology–Neonatal Intensive Care Unit, IRCCS Casa Sollievo Della Sofferenza, 71013 San Giovanni Rotondo, Italy; lauracomegna@gmail.com (L.C.); rainbow9097@hotmail.it (P.L.); ma.gentile@operapadrepio.it (M.A.G.); antonio.villani@operapadrepio.it (A.V.); 2Human Milk Bank, “Casa Sollievo Della Sofferenza” Foundation, 71013 San Giovanni Rotondo, Italy; palumbogiuseppina@tiscali.it; 3Statistical Department, “Casa Sollievo Della Sofferenza” Foundation, 71013 San Giovanni Rotondo, Italy; m.copetti@operapadrepio.it; 4Faculty of Medicine, University of San Raffaele Vita-Salute, 20132 Milano, Italy; federicazambii@gmail.com

**Keywords:** multiples’ breastfeeding, factors of breastfeeding duration, preterm twins’ breastfeeding

## Abstract

Background: Over the past year, there has been a rise in twin births. The current scientific consensus recommended breast-feed milk for all newborns for at least 6 months. They stated that it is possible to meet the nutritional needs of two or more newborns with only one mother’s milk. More information would be desirable about the factors that influence or lead to the initiation and interruption of breastfeeding. The quality of the evidence available from multiple studies has been inconclusive and therefore led to controversial interpretations and practices. Aims: The first aim of this study was to analyze the extent of the feeding of multiples with breast milk in the experience of our clinical unit in terms of incidence and duration. The second objective was to evaluate the correlation between maternal, perinatal and neonatal variables with breast milk feeding rates and duration. Methods: The study was conducted between 2015 and 2020, in a NICU in Southern Italy (San Giovanni Rotondo, Foggia). Sixty-one women who have given birth to multiples were enrolled into the study. Newborn data were retrospectively collected by informatic database and breastfeeding information were collected by a questionnaire. Results: In our centre, the percentage of twins out of the total number of births over the years has almost doubled from 1.28% in 2015 to 2.48% in 2020 and the 88% of twins are premature. 18.1% received breast milk for more than 6 months and 6.3% received it for more than 12 months. Infants of lower gestational age and weight, born to multiparous, more mature and medium-high schooling mothers received breast milk for a longer period. 35% of women explained that the interruption of breastfeeding was due to the insufficient milk production and 41% to the stress and difficulties in managing the twins. Qualitative analysis of maternal narrative revealed, for many of them, the awareness of the importance of breastfeeding and the efforts made to try to give breast milk, but also fears about the quantity of milk and satiety of their children. Conclusions: It is important to identify the factors both favoring and obstructing maternal milk feeding of multiples and it would be desirable the activation of a network of training and support for mothers after discharge, with particular regard to the categories found to be less inclined.

## 1. Introduction

Based on a recent study [1], the highest peak of twin births in the history of humanity has been recorded in the last years; now, one out of every 42 births is a twin birth. Since the 1980s, the global twinning rate has increased of one-third, from 9.1 to 12.0 per 1000 deliveries, with about 1.6 million twin pairs born each year.

In Italy, in the last twenty years, multiple births increased by approximately 25% as a result of two factors: the rising age of new mothers and medically assisted procreation.

Furthermore, 6.6% of newborns are born before the 37th week, and a quarter are twin births, for which the likelihood of low birth weight is ten times greater. The increasing number of multiple births is related to increased rates of preterm birth and low birthweight [2].

The feeding of preterm twins is an important aspect, but the attention and data relating to this issue are limited.

Exclusive feeding with breast milk is indicated for all newborns for at least 6 months [3,4] due to its unique beneficial properties [5,6]; for premature twins, human milk is important for its demonstrated protective effect against the main complications of prematurity [7,8,9] and for its role in neurocognitive development [10,11,12,13,14].

In our experience, the belief that the mother’s milk is not enough for two babies is still present, even though the WHO and other authorities state that it is sufficient for feeding multiple babies [15].

The first aim of this study was to analyse the extent of the feeding of multiples with breast milk in the experience of our clinical unit in terms of incidence and duration.

The second objective was to evaluate the correlation between maternal, perinatal, and neonatal variables and breast milk feeding rates and duration.

## 2. Methods

### 2.1. Study Design

Twins’ birth data (delivery time, parity, birth weight) were obtained using the data collection systems present at NICU (Neocare—NICU, SISWEB-regional informatic system). Mothers’ information, like socio-demographic data (age, occupation, income, and educational level), were obtained from direct contact with the twins’ mothers through interviews using a questionnaire. Information about the use of DM (donated milk) was obtained from specific database (HMB Human Milk Bank database).

The mothers of the twins were contacted by telephone for the proposal, the oral consent to the study participation, and the indications on the completion of the questionnaire. After that, a questionnaire was sent by e-mail. In some cases, more telephone conversations followed. Feedback was still positive.

### 2.2. Setting

The study was performed in the Neonatal Intensive Care Unit (NICU), Casa Sollievo della Sofferenza Foundation (CSS), San Giovanni Rotondo. They enrolled women who had a twin birth between 2015 and 2020, with the exclusion of those born with diseases incompatible with lactation, newborns transferred to other centers, or those deceased. In the hospital, there has been a Human Milk Bank since 2010, and all babies weighing less than 1800 g receive donor human milk before their mother’s milk becomes available. All mothers of newborns who are transferred to NICU immediately after childbirth are equipped with a breast pump and follow a breast stimulation protocol every three hours.

When the enteral nutrition reaches 60 mL/kg, the human milk (maternal or donor milk) is fortified, and the feeding with formula milk begins only in the absence of breast milk at the moment of discharge from NICU. For healthy, full-term infants, the first latch on the breast occurs within few minutes after birth while also doing skin-to-skin contact, then the newborn remains close to the mother for the first two hours and in the same room the following days (rooming in 24 h).

### 2.3. Study Participants

We enrolled mothers who gave birth to multiple babies at the “Casa Sollievo della Sofferenza” in the past 6 years. These women were called for preliminary information. Eighty-two women, mothers of 168 twins, were contacted by telephone, and 64 of them (132 twins) agreed to receive a questionnaire and 61 (127 twins) completed it. (Figure 1).

### 2.4. Ethics and Data Collection

The questionnaire contained 10 questions (Figure 2). The meaning of the study was explained to each woman after oral consent.

In summary, the women were asked if they had breastfed their twins, the duration of feeding with breast milk, and if the feeding was exclusive or complementary. They were also asked the reason for the interruption and if they had received help in managing the twins and from whom. In the case of twins born prematurely, we asked if they had received donated milk or if they had donated it (Figure 2).

All the data obtained from the various data collection systems were recorded in an internal dedicated database. The data were considered for each individual twin.

In each of the 127 records, maternal data (age, educational qualification, profession, nationality), pregnancy and perinatal data (spontaneous or assisted reproductive technology, parity, number of twins, type of birth), neonatal data (birth weight, gestational age, order of birth), and the schematized data of the answers to the questionnaire were collected and reported longitudinally.

### 2.5. Data Analysis

The data were processed as population studies, and descriptive statistics were used to outline and summarize mother and infant characteristics. 

Demographical, clinical, and feeding subjects’ characteristics were reported as mean and standard deviation (or median and range) and as frequency and percentage for continuous and categorical variables, respectively.

Group comparisons were performed using Pearson chi-square test for categorical variables and Mann–Whitney U test or Kruskal–Wallis test, as appropriate, for continuous variables. 

A *p*-value < 0.05 was considered as statistically significant. All analyses were performed using R (version 4.0.2).

## 3. Results

The percentage of twins out of the total number of births over the years has doubled from 1.28% in 2015 to 2.48% in 2020 (Table 1)

### 3.1. Population

Sixty-eight (54.4%) twin babies were first children, 39 (31.2%) came after another pregnancy, 16 (12.8%) after two previous pregnancies, and 2 (1.6%) after three (Table 2 and Table 3).

114 (89.8%) newborns were born from bigeminal birth, 9 (7.1%) from triplet birth, and 4 (3.1%) from quadruplet birth. (Table 2 and Table 3)

67.6% of the births, where the data are available, come from medically assisted pregnancy; 90% of newborns were born from CS (Table 4, Figure 3).

Ten (7.87%) newborns weighed < 1000 g (ELBW), 29 (22.8%) < 1500 g (VLBW), and 100 (78.7%) < 2500 g (Table 5, Figure 4). 

The gestational age for the multiples ranged between 25 and 37 weeks. 111 of 127 (87%) were premature infants: 40 twins (31.4%) were born at ≤32 weeks of gestational age and 16 (12.5%) at 37 weeks. (Table 5, Figure 5)

The age range of the mothers was 22 to 46 years. A 44.9% were under 36, and 32 (55.1%) were over 36. Statistical comparison was made between these two age categories (Table 6). 

In our cohort of mothers with twins, there were 49.5% that obtained a high school diploma, 29.7% a university degree, 20.7% secondary school leaving qualifications, and for 16 women, the data were missing. Three women are Romanian, and all the others are Italian (Table 3).

### 3.2. Breastfeeding and Use of MM

Ninety-six (75.5%) twins received breast milk, and 31 (24.4%) did not.

Seventy-nine (62.2%) twin babies were breastfed, and 48 (37.8%) were not (Table 4 and Figure 3).

Infants fed with MM > 6 months: 23 (18.11%); infants fed with MM > 12 months: 8 (6.3%); infants fed with MME > 6 months: 12 (9.44%); infants fed with MME > 12 months: 8 (6.3%) (Table 3 and Table 7).

Forty-one newborns has a preterm GA (gestational age) < 32 weeks; 32 (78%) received MM for a period ranging from 2 months and 36 months. Twenty-six (63.4%) were breastfed. Nine (22%) did not receive MM. The duration of the EMM (exclusive maternal milk) ranged between 1 month and 36 months. Out of 10 infants below 1000 g, eight were breastfed and were fed with MM for a period ranged between 3 months and 36 months. Nineteen out 29 VLBW were breastfed, and 22 (75.8%) received MM for a time ranging from 2 to 36 months. The duration of EMM feeding ranged between 1 month and 36 months (Table 3 and Table 7).

The statistical comparison Tables: education, birth weight 1,2, gestational Age 1,2, MAP, maternal age 1,2, nationality, parity, profession, type of birth) examined the relationship between the different variables and:the use of the MMthe breastfeedingthe duration of feeding with MMthe duration of feeding with exclusive MM.

The following associations were statistically significant:-The type of profession, the duration of breastfeeding, and exclusive feeding with MM were correlated, with better performance for the workers;-The type of schooling, the use of the MM, and the total duration of feeding with MM were correlated: the use of MM and its duration were higher for graduates;-Parity and total duration of the use of the MM were correlated with better performance for the pluriparous;-Maternal age was correlated with the duration of feeding with MM and feeding MM exclusively, in favour of the age category > 36 years;-Birth weight was correlated with the duration of breast feeding and of feeding with exclusive MM: we have seen better results for the infants born weighing < 1750 g;-The type of birth was correlated with the duration of feeding with MM and of feeding exclusively MM which was statistically higher in those born from spontaneous birth (even considering that 90% of twins were born by Caesarean section).

### 3.3. Reasons for Not Breastfeeding and Availability of Help

When asked about the reason for the interruption, 35% replied little milk, 41% stress and difficulties in management (other young children, effort in pumping, anxiety), 17.6% maternal diseases (sclerosis multiple, hypertension, diabetes, stress, dermatitis), 5.9% paediatric hospitalization, and 5.9% maternal will.

However, the main reason for starting formula milk was the extreme concern about the babies being not completely sated.

From the narration of twins’ mothers, many of them are aware of the importance of breastfeeding and of the efforts to give their milk either by attaching them to the breast or by pumping the milk to give it with the bottle. Some women reported a feeling of regret for not insisting enough or even guilt for not giving the twins the same opportunity as their previous children. Their fears about the quantity of milk and the satiety of their children emerge from the stories. This is evidenced by a mother who stated that:


*“I have always tried to breastfeed but with little results because I didn’t have much milk and for two months I even pumped it, then I stopped with great regret; it was difficult to breastfeed two babies at the same time… the post-operative convalescence was tough”.*


We understand the sense of regret even from another mother who says that:


*“I have been giving my milk to the twins for about 3 months. Unfortunately, already in the hospital, my milk did not meet their needs and it was integrated with the milk do-nated by other mothers. Furthermore, I did not have the pleasure of feeding them to the breast because once they came out of the incubator they were used to bottles and with the help of the nurses I went back to attaching them for a while but I couldn’t do it at home. I probably had to insist more but I was insecure and afraid. This is the only thing I regret”.*


Another woman who regrets not having insisted:


*“I tried for a month helped by the formula milk but it wasn’t sufficient. I regret not having insisted more. Maybe I chose the simplest thing: formula milk. I had a two-year-old baby when the twins were born and I nursed him for 19 months”.*


Lucia complains that she did not have enough time and strength:


*“I was able to give my milk only for the first 3 months, and then it was no longer enough. I wasn’t able to pump it due to diapers and lullabies. Time and strength were few”.*


There have been examples of women who have succeeded with willpower to go be-yond their fears and difficulties.


*“With a lot of will and stubbornness I was able to breastfeed both twins exclusively at the breast. I mean, you should give more confidence to the mothers of the twins. One of the doctors was afraid I would have abandoned the breastfeeding as if the milk was not enough. We know full well that this is not the case”.*



*“I breastfed both twins for eight months. Then I had to take it off to regulate sleep because in the night, children were always attached. This helped my family a lot”.*


Less than half of the mothers, once at home, continued to pump the milk and give it with a bottle.

Regarding the question concerning the help received at home, 46.1% replied that they had not received any help; 53.9% received help from family members: 78.6% from grandparents, 14. 2% from older children and 14.2% from aunts.

The average duration of breast milk feeding was 5.45 months for the twins of mothers who had benefited from help and 2.8 months for twins born to mothers who did not receive any help at home, with a statistically positive correlation between the duration of breastfeeding and the help.

Other exceptions concerned those who did not want any help: the mothers of triplets. They tried to manage the organization exclusively between mother and father. This al-lowed them to regulate “the sleep-wake rhythm as much as possible, with the management of the cuddle by inventing the strangest things…”

No mention of help provided at home by medical staff or dedicated professional figures was made.

## 4. Analysis of the Results

### 4.1. The Frequency of Twin Births

In the population studied, the percentage of twins has almost doubled from 1.28% in 2015 to 2.48% in 2020. According to literature, the majority of countries showed a substantial increase in twinning rates except for the poorest countries, such as Africa and South Asia and a number of countries in Central and South America [1].

In the United States, multiple pregnancy rates increased from 19.3 to 30.7 out of each 1000 live births between 1980 and 1999 [16], while in England, the rate increased from 10 in each 1000 in 1980 to 16 in each 1000 in 2011 [17]. In Brazil, the incidence of twin pregnancies increased from 1.82% to 2.03% in the period from 2005–2015. European rates in 2000 varied from 12.2 in each 1000 maternities in Italy to 19.4 in each 1000 in the Netherlands [18].

In developing countries, rates between 9 and 18 per 1000 live births have been reported [19]. This rise and variation is partly due to the use of reproductive techniques and clarified by major births from older women [20]. More recently, rates of higher-order births have declined since changes in assisted reproductive techniques have been implemented in order to (ART) reduce multiple pregnancies [21,22].

### 4.2. Breastfeeding of Premature Twins

The incidence of multiple births, having risen in the last decades, proves that the proportion of infants born preterm is higher compared to those born singleton. In this study, more than 88% of twins were premature, and 32.28% were of gestational age < 32 weeks.

Multiple pregnancies increase the possibility of a range of adverse perinatal outcomes, including breastfeeding failure [23]. In this study, breastfeeding occurred for 65% of the twins, and this is encouraging, but 18.1% received MM for more than 6 months, and 9.4% received exclusive breast milk. 6.3% received MM for more than 12 months, and 6.3% received exclusive MM. Nevertheless, partial breastfeeding is better than no mother’s milk. The average length of donor milk feeding was 19.4 days, with a minimum of 2 days and a maximum of 63.

### 4.3. Maternal Factors

#### 4.3.1. Educational Level

In the study population, the level of education has a statistically significant correlation (*p*: 0.007) with a longer duration of breastfeeding among graduates and high school graduates compared to women with a middle school leaving certificate. The data contrast with the correlation found between the duration in months of breastfeeding (*p* < 0.001) and the type of profession, showing better breastfeeding data for workers compared to office workers and freelancers.

We would tend to interpret it by saying that the degree of education has more significance than the type of profession with regard to the propensity to breastfeed. It also suggests identifying lower-educated mothers as a priority group that need more support and guidance from family and professionals.

One proposed explanation for the association of duration of breastfeeding and a lower education is that high self-esteem and breastfeeding are highly connected, so a lower education could mean that the person has less status, less control, and less power, which in turn may produce stress and lower self-esteem [24,25,26,27].

#### 4.3.2. Age

The mean age of the mothers in a cross-sectional survey involving 185 mother–twin pairs [18] was 30.18 +/− 1.29 years. In our population, the average is higher: 35 years with the median of 36.

A statistically significant association was revealed between the two categories of maternal age (more or less than 36 years) both for the duration of feeding with breast milk and with exclusive breastfeeding. Older women breastfed longer.

#### 4.3.3. Birth Weight

We found a statistical correlation between birth weight and the duration of feeding with both total and exclusive breast milk, with higher percentages in the category of premature babies weighing < 1735 g.

This is a fact that must be interpreted. It confirms the effectiveness of care aimed at promoting breastfeeding for very low birth weight infants, with dedicated and standardized protocols, but also suggests that support for mothers of preterm not VLBW should be intensified.

#### 4.3.4. Type of Birth

As regards the type of birth in our sample of twin pregnancies, 90.6% were by Caesarean section. The statistical comparison was significantly in favour of those born from spontaneous birth regarding the result concerning exclusive breastfeeding with breast milk.

Early weaning in the Caesarean population is consistent in various studies [19] but not in the prospective randomized trial conducted in Brazil [28]; however, it should be noted that because higher rates of CS (Caesarean section) are observed among twin pregnancies worldwide, the rates of weaning might be biased by this high rate.

#### 4.3.5. Availability of Help

In our population, 46.1% reported that they had not received any help in the management of twins, and the remaining percentage answered yes, albeit half of them received help for a limited period. In 78.6%, the help came from grandparents. In the last year, aid has failed due to forced isolation for the pandemic, and this underlines the precious role of grandparents and the importance of this age category so tormented by the COVID-19 pandemic.

The year of the pandemic is just one of six years of observation in the study; it has been an exceptional time and hopefully is limited in time.

We consider the impact of the availability of help on the duration of breastfeeding to be very significant, with an average of 2.65 months more spent breastfeeding for twins born to a mother who received help than those born to a mother who had not received it and with the statistical significance between the two elements. This underscores the importance of family support for the duration of breastfeeding.

## 5. Discussion

All the international organizations recommend exclusively breastfeeding for the first six months of life and continuation for at least one year once other food is introduced [3]. The initial six months of life are critical for the infant’s health and development. Achieving optimal development and minimizing the risk for serious disease in the newborn is an important challenge for clinicians. High-risk infants, such as premature twins, may benefit from mothers’ milk; the duration of breastfeeding has remained below that for single, full-term infants.

The best data on the rates of twins’ breastfeeding come from Sweden, where breastfeeding frequencies in preterm twins were 79% at 2 months, 58% at 4 months, 39% at 6 months, 14% at 9 months, and 6% at 12 months. In Poland, exclusive breastfeeding among twins and triplets was 4.9%, while that of the singletons was 73.2% [22]. At six months, the overall breastfeeding rate in twins ranges from 11.1% to 49.6%, and the exclusive breastfeeding rate ranges from 4.1% to 21.5%. Another study [29] in Denver reported that at six months, about 25% of twins and 15% of triplets receive breast milk. In our population, 22.8% received breast milk at 2 months, 7.9% at 6 months, and 4.7% at 12 months. This percentage is insufficient compared to the recommended standards and suggests that we should identify the factors for which we could intervene to improve it.

Previous researches [30,31,32,33,34,35,36,37] have identified associations between breastfeeding in the NICU and maternal characteristics, infant characteristics, and environmental factors. Many studies [38,39,40,41,42,43] reported the risk factors for early interruption of breastfeeding in mothers of singletons, and the most common are young age, low education, and single marital status.

Findings from a Swedish study [19] indicate that maternal factors, such as educational level, smoking, and maternal age, were all individually related to the interruption of breastfeeding before six months of age, with low maternal educational level being an especially important factor.

In 2018, it was reported that there was a statistically significant impact of maternal age and profession on the volume of milk donation [38].

In this study, the correlation between the variables education, maternal age, parity, profession, type of birth, birth weight, gestational age, and the months of breastfeeding was statistically significant.

The presence of other children served as a stimulus for the mothers who wanted to give the twins the same health opportunity as the other children, or the older siblings were of help in the management of younger twins. For others, it represented a cause of stress and regret for not having been able to breastfeed for the same duration as for the previous child.

Women who have previous breastfeeding experience might be more prepared to breastfeed in a subsequent pregnancy and feel safer starting and maintaining breastfeeding two infants.


*“I have been nursing the twins for about three months. I had to stop because my two other children cried. They always wanted to be with me. As you well know, you need to be tranquil if you breastfeed because it is a time you want to dedicate exclusively to your newborn. I missed this as I found myself with four small children together. I was very sorry but I had to make a choice: they would have risked having an exhausted mother!”*


In the practice of the human milk bank, multiparous women were more numerous than primiparous ones [38].

The Swedish study [19] indicated that term infants had longer breastfeeding duration if the mother was multiparous, in agreement with other studies on the impact of parity on breastfeeding duration [22,33]. However, there is an opposite outcome [23,26] if the infants are born preterm; then, being multiparous may not be as beneficial.

This is because multiparous mothers of preterm twins may experience a greater deal of exhaustion with older children at home and newborns admitted to a neonatal unit [39], whereas primiparous mothers of preterm twins may have more time for the infants and can be focused on breastfeeding.

Our results suggest intensifying care aimed at promoting breastfeeding for primiparous mothers.

Prematurity adds to the already significant challenges of initiating and maintaining lactation for twins and higher order multiples [23]. Preterm twins showed an increased risk of being weaned compared to term [19] and twin infants born prematurely, who are less likely than twins born at term to be breastfed.

Some studies [40,41,42,43] reported that higher gestational age is associated with higher rates of breastfeeding.

The data are, however, contradictory since some authors [43] reported that 70% of mothers did not practice exclusive breastfeeding as a result of NICU admission, while others [44] found 68% of mothers practiced exclusive breastfeeding at NICU.

In our study, there was a higher incidence of exclusive breast milk feeding among multiples with preterm gestational age < 33 weeks. It is further confirmation of another work [45] where we highlighted that, if motivated and if breast stimulation is initiated early on, mothers of extremely premature infants can exclusively feed their infants in NICU with breast milk. This is in agreement with others studies [24] that have found that mothers of VLBW infants acknowledge that human milk is a unique factor in improving outcomes for their infant, and consequently, breastfeeding rates in such cases are higher.

Probably, as for VLBW, gestational age-specific breastfeeding promotion protocols should also be intensified for twins of greater GA.

Older women breastfeed longer. This advocates the need for greater support for breastfeeding twins for younger women.

In this analysis, the main reason for interrupting breastfeeding was insufficient milk production.

Studies confirmed that the most common reason given for the interruption of breastfeeding by mothers of multiples [46] is insufficient milk production. Mothers may associate infants’ excessive crying or irritability with symptoms of hunger and, consequently, with insufficient human milk supply [28]. It is estimated [47] that six months after childbirth, the body of a mother of twins can produce from 1.0 to 2.0 kg of milk per day, while a body of a mother of triplets can produce more than 3.0 kg. Instead, in our experience, the idea of insufficient milk production for two or more newborns is considered a common opinion of many mothers and caregivers. Nevertheless, it has no scientific support.

One of the lessons that comes from this work is that communicative effort must be intensified to convey the message of the adequacy of the amount of milk that the mothers of the twins can produce.

Some authors [23] are convinced that the gap between breastfeeding of twins and that of singletons and that between the percentages of onset and duration may be reduced by evidence-based interventions tailored to families with multiples. Mothers of twins may benefit from breastfeeding support approaches tailored to their needs.

In addition to the family context, educated healthcare professionals should give relevant information about the adequacy criteria for mother’s milk and encourage the mothers in breastfeeding during the early stages of the establishment of breastfeeding. Since in the area there are no experts who can provide breastfeeding assistance at home, illustrated documents and videos about nourishment and nursing for the multiple babies should be offered to the mothers.

Another proposal that is being planned is the activation of a telephone number dedicated to breastfeeding: each mother who needs advice, verbal help, comfort, or reassurance on issues concerning breastfeeding can contact the nursery and find a doctor or nurse who is ready to provide the necessary support.

From the qualitative analysis of the answers, it emerged that the second deterrent to the prolongation of breastfeeding is the stress associated with managing two or more babies.

The women who had help available received it from family members. In our study, those who received help breastfed the longest. Nevertheless, a certain percentage of women stopped breastfeeding despite having the help of family members. This is because they felt unprepared to breastfeed two babies at the same time.

Correct training of pregnant women may play a positive role with multiple pregnancies. This training must see the necessary involvement of the family, which must be material and motivational. Sometimes the invitation to abandon breastfeeding in favour of formula milk comes from family members because they are convinced that, in a way, the stress of managing twins can be reduced, while on the contrary, breastfeeding, if fully utilized, is a method of reducing stress for the mother and the little ones. Exploring the topic of possible methods of reducing stress after a twin birth and their possible factors can be a way to extend the duration of breastfeeding at least up to six months, the same as for single-born babies.

In our area, there is little presence of breastfeeding experts who, on the other hand, would be crucial for the goal of improving breastfeeding for twins. The ideal would be that of a multidisciplinary team that supports mother and family in dealing with the clinical, psychological, and practical management difficulties in the weeks following the birth from the moment of discharge.

In the experience of the women interviewed, it emerged that there is a lack of proper breastfeeding preparation during pregnancy and an absence of expert voices after returning home.

The keywords of the approach to pregnancy and childbirth are information and support. We would like these to become a mantra for the care of women who live this intense experience of life that is the birth of twins. This could lead to greater personal conviction.

In our experience, mothers who have breastfed for more than 12 months are those who initially had a strong personal belief that breast milk was a unique opportunity for their children and strongly believed in the power of breastfeeding.

Women with a stronger motivation gave priority to breastfeeding in the management of twins, and this has led to brilliant results in terms of breastfeeding duration.


*“I wanted to pump my milk immediately after the birth of my 900 g twins, be-cause I only wanted to give my milk and, as soon as it was possible, I attached them to the breast. They still are and are 4 years old. Then I felt sorry for the other babies and I donated my milk to them too”.*


In another interview:


*“I breastfed both twins until the twentieth month of life then I voluntarily decided to stop even if I still had milk. The breastfeeding experience of the twins is unique, I often used the rugby position to please both at the same time even though I much preferred exclusive breastfeeding with each one to fully enjoy that intimate and unrepeatable moment”.*


Did you have any help with the twins?


*“Yes, of course, the two grandmothers”*


A weaker motivation, despite the awareness of the benefits of breast milk, often gives way to fears of not being able to feed the twins with their own milk or manage family life. This context triggered regrets and guilt and led to the abandonment of breastfeeding in the first 2–3 months. Women who were convinced that enough milk could not be produced for more babies abandoned breastfeeding early.

When a mother is pregnant with twins (or multiples), a screening to select whether a mother needs a low, moderate, or higher level of support might be useful.

### Limits and Opportunities

One of the limits of this study is represented by the fact that the data came from a single centre, while a survey on the whole national territory would be useful. The valuable information resulting from this experience could represents a useful opportunity for improving the breastfeeding rate of twins.

## 6. Conclusions

The incidence in our clinical centre of twin births doubled in six years, in line with Italian and European data. The use of breast milk and the breastfeeding of twins recorded percentages below the standards indicated by the WHO. The age, the educational level, and the parity were the maternal factors statistically correlated with the duration of breastfeeding. Better results were obtained with older mothers, a higher educational level, and multiple parity. These data suggest the desirability of greater support for mothers who are younger and with a lower educational level. Multiparity, on the other hand, did not represent a prejudice to prolonged breastfeeding.

The neonatal factors that impacted breastfeeding rates were birth weight and gestational age, with the best performance observed for the lower weight and lower gestational age categories of newborns. Care dedicated to breastfeeding should be intensified for mothers of siblings of greater weight and gestational age.

The main reasons for interrupting breastfeeding are insufficient milk production, stress and difficulties in managing the twins. The concept that breast milk may be sufficient to breastfeed twins should be more widespread, as reiterated by the WHO.

About 50% of mothers of twins did not receive any help. The presence of experts for the support of mothers was not even named in our results, confirming a welfare gap in the area. Support was provided within the family, and the most important role was played by grandparents, who are traditionally fundamental social figures, particularly in this historical period in our area. Family support had a statistically significant impact on the duration of breastfeeding. Support implementation programs run by health professionals in the first months after delivery are desirable for stress containment.

Qualitative analysis revealed that, for many mothers, there was an awareness of the importance of breastfeeding and efforts made to try to give breast milk, but there were also fears about the quantity of milk and satiety of their children.

It would be interesting to evaluate the outcome of twins fed with breast milk and those with formula milk, and this will be the subject of a future study.

## Figures and Tables

**Figure 1 nutrients-13-03099-f001:**
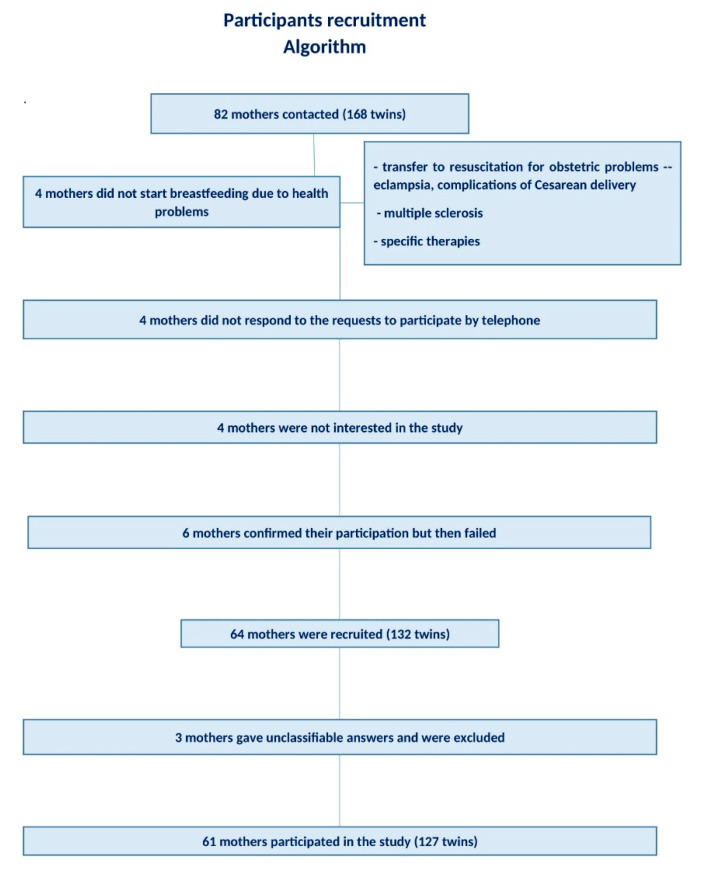
Algorithm partecipants recruitment.

**Figure 2 nutrients-13-03099-f002:**
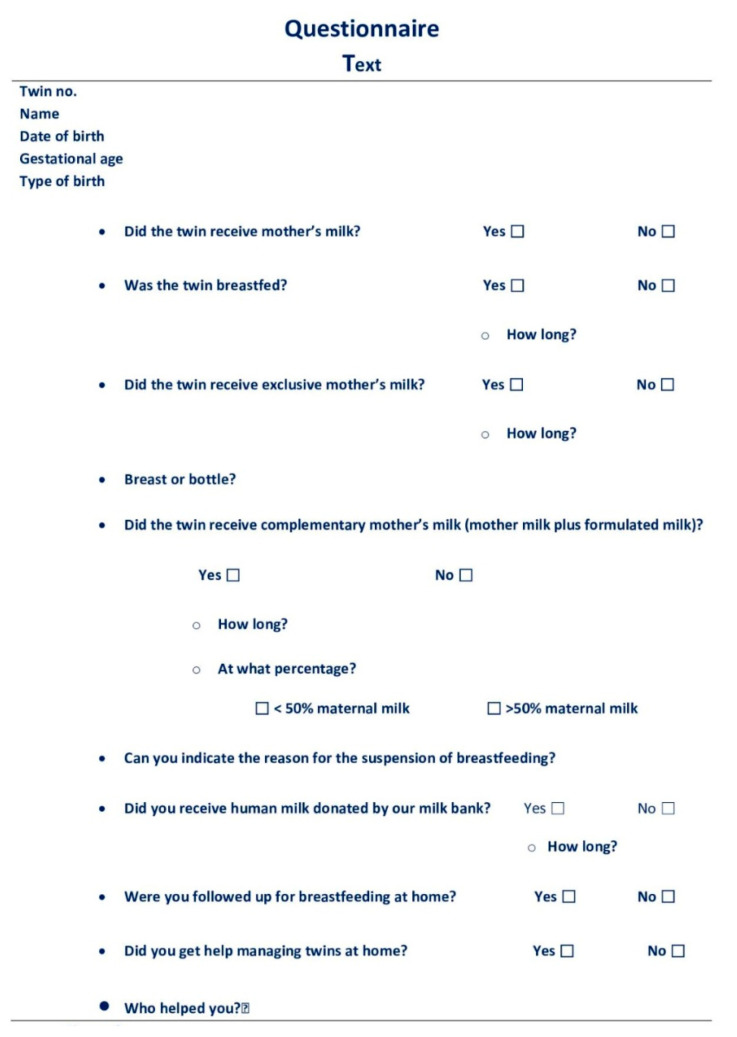
Questionnaire Text.

**Figure 3 nutrients-13-03099-f003:**
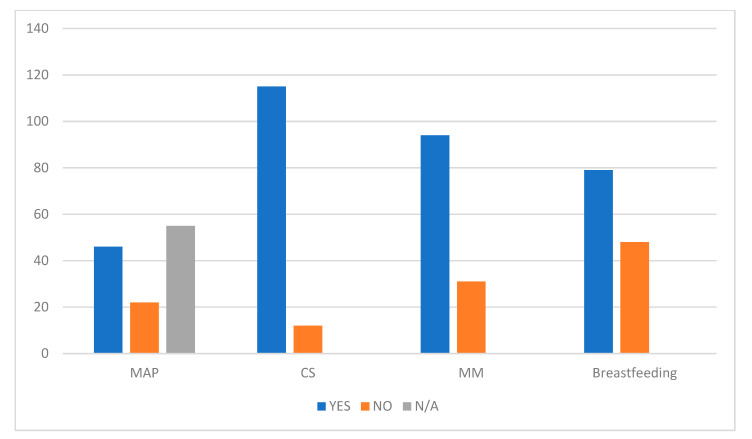
MAP, medically assisted procreation; CS, Caesarean section; MM, maternal milk.

**Figure 4 nutrients-13-03099-f004:**
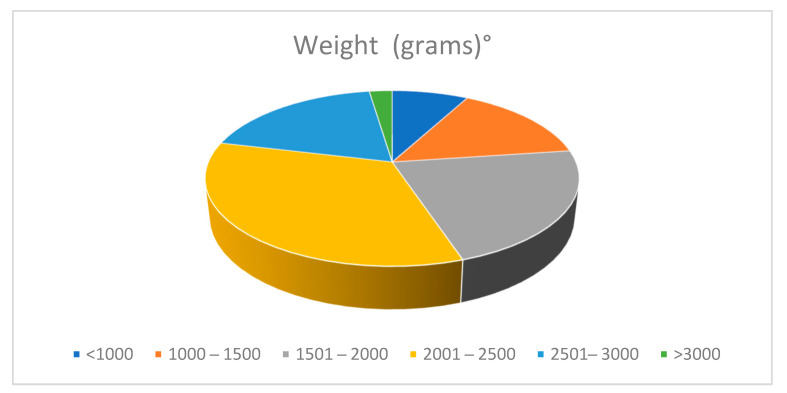
Population data. Distribution by weight.

**Figure 5 nutrients-13-03099-f005:**
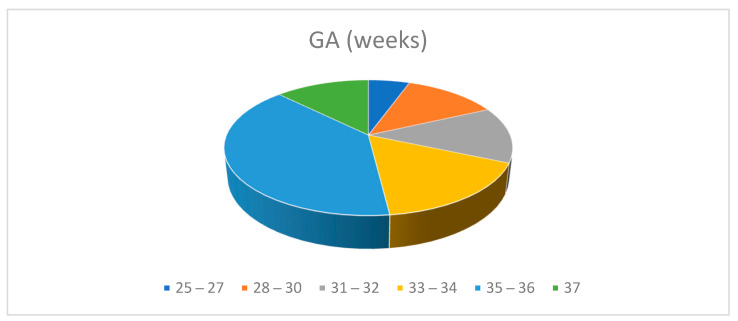
Population data. Distribution by Gestational Age.

**Table 1 nutrients-13-03099-t001:** Annual incidence of twin births.

2015	2016	2017	2018	2019	2020	2021
1.28%	1.32%	1.6%	1.83%	2.29%	2.48%	2.4%

**Table 2 nutrients-13-03099-t002:** Parity and n° twins.

	1	2	3	4
Parity	68 (54.4%)	39 (31.2%)	16 (12.8%)	2 (1.6%)
Number of newborns per delivery	/	57 (93.4%)	3 (4.9%)	1 (1.63%)

**Table 3 nutrients-13-03099-t003:** Mean, median, and range.

	N°	%	Mean (SD)	Median (Q1, Q3)	Min–Max
**Maternal age**			35.12 (5.57)	36.00 (31.00, 3900)	22.00–46.00
**Parity**					
1	68	54.4			
2	39	31.2			
3	16	12.8			
4	2	1.6			
**Educational level**					
Middle School	23	20.7			
Diploma	55	49.5			
Degree	33	29.7			
**Nationality**					
Italy	98	94.2			
Romania	6	5.8			
**Profession**					
Housewife	53	41.7			
Employee	25	19.7			
Self employed	16	12.6			
Worker	19	14.9			
Unemployed	8	6.3			
**N° twins**					
2	114	89.8			
3	9	7.1			
4	4	3.1			
**Pregnancy type**					
Assisted Fertilization	46	67.6			
Spontaneous	22	32.4			
Missing N	59				
**Gestational age**			33.57 (3.00)	35.00 (32.00, 36.00)	25.00–37.00
**Birth weight**			2010.84 (606.73)	2080.00 (1605.00, 2475)	750.00–3260.00
**DM days**			19.4 (17.85)	12.00 (5.00, 27.00)	2.00–63.00
**MM**					
Yes	94	75.2			
No	31	24.8			
**Breastfeeding**					
Yes	79	62.2			
No	48	37.8			
Months			3.50 (5.32)	2.00 (1.00, 4.00)	0.00–36.00
**EMM months**			2.31 (5.28)	1.00 (0.00, 2.00)	0.00–36.00
**CMM months**			1.16 (1.54)	1.00 (0.00,1.00)	0.00–6.00

DM, donor milk, MM, maternal milk, EMM, exclusive mother milk, CMM, complementary mother milk.

**Table 4 nutrients-13-03099-t004:** MAP, CS, Feeding with MM, Breastfeeding.

	MAP	%	CS	%	MM	%	Breastfeeding	%
YES	46	67.6	115	90.5	94	75.2	79	62.2
NO	22	32.4	12	9.4	31	24.8	48	37.8
	68							
N/A	55	43.3						

MAP, medically assisted procreation; CS, Caesarean section; MM, maternal milk.

**Table 5 nutrients-13-03099-t005:** Weight and Gestational Age.

WEIGHT						
**Grams**	**<1000**	**1000–1500**	**1501–2000**	**2001–2500**	**2501–3000**	**>3000**
N°	10	19	28	43	24	3
%	7.87	14.9	22	33.8	18.8	2.36
**GESTATIONAL AGE**						
**Weeks**	**25–27**	**28–30**	**31–32**	**33–34**	**35–36**	**37**
N°	7	16	17	21	50	16
%	5.5	12.5	13.4	16.5	39.3	12.6

**Table 6 nutrients-13-03099-t006:** Maternal Age.

	N°	%	
<=30	10	17.2	
31–35	16	27.5	
36–40	20	34.5	55.1%
>40	12	20.7	

**Table 7 nutrients-13-03099-t007:** Breastfeeding, MM feeding and EMM feeding of the preterms.

	N°	Breastfeeding	MM Feeding	EMM Feeding
<32 weeks GA	41	26 (63.4%)	32 (78%)	25 (61%)
VLBW	29	19 (65.5%)	22 (75.8%)	16 (55.1%)
ELBW	10	8 (80%)	8 (80%)	8 (80%)

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
