# Peer review of "Feeding Twins with Human Milk and Factors Associated with Its Duration: A Qualitative and Quantitative Study in Southern Italy"

_nutrients, 2021, doi:10.3390/nu13093099_

Round 1
Reviewer 1 Report
Review of paper entitled: "Twins breastfeeding and use of human milk. Factors associated with duration".
This is a mixed quantitative and qualitative analysis of 127 newborns and their mothers recruited over a span of 6 years (inclusive of the last Covid -related year), and where their trend of breastfeeding duration was assessed in correlation to maternal, perinatal and neonatal influencing factors.
It is an interesting study that covers the description of a local population within one of the regions of Italy and gives a narrative slice of the lack of health care organization in the region of Puglia (southern Italy). Such study could be analyzed and compared by colleagues in similar regions and provinces.
Nevertheless, in order to allow colleagues and other researcher to fully benefit of this analysis, and ensure reproducibility, I would request a complete rework as follows point by point.
Title – Conveys unclear meaning. The use of breastfeeding and use of human milk in the same phrase is redundant if not confusing. Suggest revising title and make it a single phrase. I would also include in the title the fact that is a local, single experience within one of the Italian regions.
A. Abstract – needs to be majorly reworded and clarified
Background –
- “The possibility of being fed with breast milk is completely insufficient compared to the breastfeeding rates recommended by the WHO and this further impacts on the outcome” – the message is not clear and the terminology like “completely insufficient” is a blunt statement that could be replaced by something like: “…might not be possible for all as compared to…”
- “ The available data on the factors that influence or lead to the initiation and interruption of breastfeeding are rare and conflicting” – By looking at your reference list there are at least 36 - if not more articles- that tackles the issues of different factors influencing the breastfeeding when women have multiples. And even if I can agree with the statement that some of the factors can be controversial, this topic has been well described in the literature and many more articles are missing to your list, which does not justify the erroneous assumption that this topic has been rarely described.
- O Flidel-Rimon, E S Shinwell. Breast feeding twins and high multiples. Arch Dis Child Fetal Neonatal Ed 2006;91:F377–F380. doi: 10.1136/adc.2005.082305 and many others published reviews;
- Arora S, McJunkin C, Wehrer J, et al. Major factors influencing breastfeeding rates: mother’s perception of father’s attitude and milk supply. Pediatrics 2000;106:e67.
- Kim, BY. Factors that influence early breastfeeding of singletons and twins in Korea: a retrospective study. Int Breastfeed J12, 4 (2016). https://doi.org/10.1186/s13006-016-0094-5
Study – needs clear aim stated at the beginning of the phrase not in the middle.
- Totally…” – should be total.
- “127 subjects born since 2015 included in the study” – Should give an exact range of time from 2015. Was the study carried without interruptions through the year 2020 or included part of the year 2021? Please describe precisely.
Results – in general needs to be more concise and without redundant wording.
- ‘In our sample….’ – should be replaced by a fact: ‘The percentage of the twin population in our sample compared to the total number of births…’
- “more mature and medium-high schooling mothers…” – meaning not clear.
- Abbreviations like: EMM, VLBW, MM, FM, MME, MAP, GA etc. – Please do not use acronyms, please ‘spelled them out’ for the readers the first time they appear in the text of the manuscript.
Conclusion - Needs a major rearrangement and construction. Overall pro and cons factors influencing breastfeeding are already well known and studied. I would point out what your Italian local region study has contributed to analyze the population issue and represent an interesting information that could help the local or national awareness and support to improve the rate of breastfeeding of women delivering twins. It would be interesting when the authors are speaking about their territory they should give an idea about how many babies are born prematurely compared to other territories within the same Italian region; to have an idea not only of the proportions but also about the impact they may have Needs to be redone completely.
- “to counter the widespread mistaken belief, denied by the WHO, that it is impossible to meet the nutritional needs of two or more newborns only with mother's milk” – could be definitely better phrased to lessen bias.
- Please replace the word ‘mistaken belief’ with
B. Introduction
- “since multiple pregnancy and birth increased the risk of adverse perinatal and postnatal outcomes” – please provide a reference.
- “The most widespread knowledge is that one mother's milk is not enough for two babies even though the WHO and other authorities state that it is sufficient for feeding multiple babies [19]. “ – could be better phrased to lessen bias. Maybe this is very strong peculiar belief in their Italian region? The authors must explain it up front, especially if it is the experience that they have noticed in their practice. The reader needs to understand where this ‘knowledge’ comes from! Who has the ‘most knowledge’? Maybe is not knowledge, maybe the authors mean a myth? Please clarify accordingly.
- “Few and conflicting studies have evaluated the breastfeeding rates and factors related to weaning in twins. Our results appear in some ways encouraging, in others disappointing, but they represent a very important source of suggestions, with the reporting of particular cases worthy of being known.” – this whole paragraph should be in the conclusion section of the manuscript.
C. Methods – I would divide the methods as follows in a very structured way: a) Study design and Settings; b) Participants; c) Ethics and data collection; d) Data analysis.
The majority of the data gathered information could be easily assembled into a table format (i.e. newborns and maternal demographics, exclusion criteria), with the aid of an algorithm you can describe about the 127 subjects, the total numbers of recruited babies at the beginning, the one lost to recruitment (for exclusion criteria or other reasons and so on), the one lost to follow-up, the one non-responding to questionnaire etc. Try to explain everything narratively is too long and confusing for the reader. A comprehensive and well organized table will give justice to the study, and would be easier for the authors to describe it.
2.1. Study Design –
- The authors switch from“127 individuals”, then describe “127 twins”, then “the mothers of 127 twins were contacted by telephone…”: so who are the subjects of your study? I would say both the newborn for quantitative analysis and the mothers that are 63.5(?) adult subjects for your qualitative/quantitative study. You need better explain the number of the pediatric and adult subjects participating to your research.
- In the first paragraph the authors explain the study as a “direct contact with the twins’ mothers” – What does it means?
- Please immediately explain that it was a questionnaire, as I see it later on the 2 paragraph another related piece of information: “…qualitative data obtained through longitudinal interviews using a semi-structured questionnaire and the extrapolation of socio demographic, biomedical data….three specific database”, lastly I found again the study method in the Data collection section as follows: “The interview to the mothers of the twins was held on the phone and the questionnaire contained 10 questions.” – It is way too scattered and it would be better to follow my suggestion on how to present your data. Were the interviews recorded and transcribed? Who reviewed the questionnaire to be sure that was easy to understand etc?
- The whole paragraph: “The main focus was on the breastfeeding of twins…in order to assess the changes of recent years” should go either in a section called aim of the study or in the discussion, as opinions and experiences can enrich that part of the manuscript.
- When the author talk about the ‘three specific database” they are meaning local, regional, national or else? Not clear please specify.
- The semi-structured questionnaire (Tab 2) is not provided for reviewer’s analysis in the pdf article; and it is rather described by the authors, as a survey with a yes or no answers. The description of a longitudinal phone call with the questionnaire does not explain the true content of the interview (are they asking open questions during their interviews?) and/or an interview format in which either transcripts of comments from the targeted population is then judged for credibility. How did the researcher reach a thematic consensus? The examples of verbatim that is possibly part of the qualitative interview with the mothers, should be better introduced, interpreted to the main goal of this study, and more examples would be probably appreciated by the readers.
2.2. Setting –
- “disease incompatible with lactation” – please mention examples in the exclusion criteria as I have mentioned above, in the overall comment.
- The word ‘bank milk’ can alternatively be called ‘human donor milk’.
- “For healthy full-term infants, the first breast attack occurs a few minutes”…please change to ‘the first latch on the breast, occurs in few minutes after birth, while doing skin-to-skin with the mother for the first two hours of life…”
- “Nevertheless, the situation of newborns’ discharge is critical in our region. In actual fact, this territory is totally lacking in professional figures dedicated to breastfeeding other than psychological and material support of women who have just given birth. It is known that the first weeks after childbirth are particularly difficult for mothers in a particular physical and psychological stress and poor hormonal balance where the only help, when available, falls to family members….to twin pregnancies” – The English structure should be reviewed, and the whole paragraph should be moved to the discussion section.
2.3. Sample –
- The title to this methodology section ‘Sample’ should be replaced by Study participants – My suggestion here is that it would be better explained to the readers as a schematic algorithm that includes the total number of recruited individuals (newborns and mothers), the numbers of subjects excluded with exclusion criteria well stated. Who participated to the questionnaire/telephonic interview, how many lost in follow up etc.
- “6-year period between 2015 and 2020/21.” – does the period end in 2021? Authors should disclose the exact period of time included in their recruitment of subjects with twin pregnancy.
2.4. Definitions –
- One comment on the way to breastfeed: what about twins that were bottle fed? Or half breastfed and ½ bottle-fed? Would other caregivers help out? Other human milk than mother? Could have this change the overall result?
- The word trio has been mentioned in the results of the manuscript – not clear if it is referred to the trio mother & twins or 3 newborns, please explain.
2.5. Data collection –
- Definitely the authors should provide a picture of the questionnaire with the 10 questions, and then just refer to it within the text in lieu of the narrative. Maybe was contained in referred Table 2. If the questionnaire is already included in Table 2, please ignore this comment.
2.6. Data analysis –
1. only the quantitative data analysis seems described, while in the methods a qualitative research was mentioned. Going back to the data collection and in connection with this point, and because the table 2 describing the questionnaire (?), is not fully available, it seems that questions were not open, but just a yes/no answer, which matches with a survey rather than a recorded narrative interviews? It would be nice to know after the yes/no response, if at the end of each questions a space for free comments?
D. Results – In the pdf copy of the manuscript the 5 tables summarizing the data results were not included. The narrative attempt of describing the data is confusing, redundant and difficult to follow because of lack of statistical comparisons when looking at the results. It ends up in a list of numbers and percentages that dilutes in details the interpretation of it all.
- This section is divided in 3 parts: a) population: very confusing when describing twins babies instead of number of ‘sets of twins’. – Please stick with one term, and eventually add it in your Definitions section; b) Use and duration of MM; c) Reasons for not breast feeding and availability of help: the last one it is a very skimpy section. This is surprising because it reflects the qualitative part of the whole study, please expand.
- There are sentences that mix babies with mothers demographic and are difficult to extrapolate. Example: “55 newborns have mothers with high school diplomas (49.5%), 33 (29.7%) with university degrees…” and “The mothers of 53 newborns are housewives, 25 are employed, 16 are self-employed, 19 workers and 8 are unemployed.” And again “The mothers of 98 newborns are Italian (94.2%) and 6 (5.8%) come from Romania”. – It would be better to explain it as follows: “In our cohort of mothers with twins, there were 49.5% that obtained the high school diploma…” and so on.
- Authors emphasize in the title and intro the importance of breast feeding as it pertains to the twin cohort, yet, in the description of their population results there are described triplets, and quadruplets, namely “ The twin births are 114 (89.8%), the triplets 9 (7.1%), and the quadruplets 4 (3.1%).” ???
- “The category of premature babies” – Is the title of another section.? Because the GA has already been described in the results section, this part gives a confusing way and version of looking at the whole percentage of prematurity among twins, and should be eliminated.
- In the section Use and duration of MM the following sentence is incomprehensible and therefore of difficult interpretation “The couple and the 27-week of GA trio received breast milk for 36 months and 6 months, respectively, and were breastfed. Both mothers donated milk to the bank, the first up to the maximum donation term which is 6 months and donated more than 50 liters of milk and the mother of the 3 twins donated about 5 liters during the hospitalization.” – please rewrite in comprehensible English.
- The authors write “In the statistical comparison (tables education, birth weight 1,2, gestational age 1,2,…) between the different variables…” – The phrase never ends and there is an entire 10-lines paragraph that doesn’t have a real meaning. Needs to be done again.
- Much of the results covered in the Discussion section could be moved back to Results section.
E. Discussion – Summarizing, the discussion section that appears overly detailed and scattered over many mini-topics, not focused on the title (factors associated with duration) and conclusions are not sufficiently balanced, with the ultimate goal which again is vague in accordance to your initial aims. In the purpose of the study the authors wish to explore trends of breastfeeding in women with twins; therefore, to be consistent, it would be important to summarize in the conclusion if an upward trend was observed, and which are the novel factors (if any) that influences the duration of breastfeeding of twins. Overall the whole Discussion section needs to be reviewed by a professional that is comfortable with the English grammatical structure and composition. Many sentences are awkward and do not convey well their meaning. Discussion lacks structure for better readability.
1. The overall discussion has an exceptional length, the narrative part is too long and full of details that are diluting the main message of the authors.
2. Improve the use of verbatim by following a logical sequence.
3. Do not mix arguments: i.e. the incidence of twin pregnancies should be separated from the incidence of breastfeeding.
4. After every mini review of the literature, there is one brief sentence about the local study, but at one point it needs to be paraphrased and integrated into the discussion text, with particular attention to a ‘continuity’ to help the reader understand what is relevant. I would suggest 3 possible actions to be considered and chosen to improve the reading:
- 1. Please summarize the different paragraphs into one general sentence: the first two paragraphs speak about increased incidence of multiple pregnancy with too many details that can be confusing; the authors should explain that the increased rate of multiple pregnancies has been trending up in the last #of years, you can name the different references for the different countries involved in similar epidemiologic studies (UK, U.S., Italy, Japan etc), then focus on your local data to compare.
- 2. Maybe you should structure the discussion in subtitles: Incidence of multiple births; Incidence of Multiple births and breastfeeding; Prevalence and associated factors of breastfeeding twins and so on. The authors could put in order their review of the literature and then make a comparison with what they have found. It may be easier to follow.
- 3. Review of the literature and interpretation of the data section. Once restructured, and summarized with comparison with local data for every topic, this Discussion will turn into a review. If the authors choose to do so, they need to remember to put the word ‘review’ in the title. The discussion will then be only focused on answering in regard of the aim of their study and on novel observations found in their local cohort.
- “…only a few studies have focused on breastfeeding twins and, consequently few evidence based interventions have been identified that are specific to mothers of twins and higher order multiples and of preterm twins. Not only, these few studies have also produced controversial results.”; I don’t agree with the author statement that there are few studies on breastfeeding multiples. In this authors’ reference list I found at least 18 articles; it also exist national and international guidelines, written by different Pediatric Societies available on line, a Cochrane review. Just reviewing how different countries have tried to support education on breastfeeding twins, I found 10 RCT supporting educations. Lastly, published analyses about factors that benefit or delay or interrupts breastfeeding in twins it is not scarce. Therefore I would suggest the authors to rephrase the paragraph with something like: “The quality of the evidence available from multiple studies has been inconclusive and therefore led to controversial interpretations and practices.”
- For your own information, please look at Multiple Births Foundation 2011; McAndrew 2012; Bonet 2011; Yokohama, 2006, Nursan Cinar, J Health Popul Nutr. 2013; Nursan Cinar, Iran J Pediatr. 2016; just to mention a few qualitative studies on the same subject.
- The authors write: “In our experience [42], if motivated and initiated early on breast stimulation, even mothers of extremely premature infants can exclusively feed their infants…” and then again “Moreover multiparous women, probably because of their better organizational skills…” – similar statements found in the whole discussion section. The reader could wonder if these statements are the reflection of some other observations in which the authors have studied and/or compared within their studied population different modes of interventions, but nowhere in the manuscript they justify these opinions/statements appropriately. Did you consider also as a ‘high motivation’ the in vitro fertilization factor?
- Please replace the bolded word: “Other papers [55] have found no breastfeeding differences between singleton and twin preterm infants.” with ”Other authors [55] have found…”
- Results like the ones I have listed below (just few examples but there are many more), belong to the results section of the manuscript rather than the discussion section; use the discussion to compare and interpret your data.
- “In this study, with a p value of 0.015, the correlation between the variable parity and the months of breastfeeding was statistically significant, while this significance was not confirmed with reference to exclusive breastfeeding”;
- “The mean age of the mothers of a cross-sectional survey involving 185 mother-twin pairs [22] was 30.18+/-1.29 years. In our population the average is higher, 35 years with the median of 36.”;
- “Statistically significant was the association between the two categories of maternal age (more or less than 36 years) both for the duration of feeding with breast milk and with exclusive breastfeeding.”
- “There was a statistical positivity (p-value 0.028) in the correlation of gestational age with the exclusive feeding with breast milk with a higher incidence among preterm gestational age <33 weeks.”
- “In our population we must give importance to the case of the foreign woman with a strong motivation towards breastfeeding and the need to feed with her milk… subsequently complemented thanks to her conviction and determination, is also noteworthy.” – these paragraphs are commendable, but would be more suited to acknowledgements section.
- “Since mothers of multiples cannot or do not want to breastfeed for a long duration may result in a significant missed opportunity, particularly as the number of multiple births continues to grow.”: this statement alone in the context is pretty obvious, and should be completed by the authors with another sentence that gives meaning to this ‘discovery’ (how do you want to support these mothers through these changes?) or better be deleted.
Authors should ask themselves the following questions and structure their discussion upon: What do the results mean for the problem and hypothesis of our study? What are the implications (if any) for the targeted population?
F. Conclusion – Major rework is needed.
- The first sentence should be broken up to smaller sentences and grammatically edited for readability.
- The authors tried to formulate a conclusion: “The ultimate goal is to identify and share a protocol of information and support, especially for women who fall into those categories less accustomed to this practice…” – The authors should try to answer to questions like: “How do our findings add to the current literature?” or again “Any suggestions for future research?”
- “For this purpose, given the current lack of data, mostly linked to the widespread belief but denied by the WHO, of the impossibility of meeting the nutritional needs of two or more newborns with exclusive mother's milk.” – This is already appearing in the abstract (please see my comments in the abstract section above), and should be rephrased in the proper English form.
G. Limits and opportunities – This whole session should go before the conclusion.
1. The phrase “…which remains disappointing all over the world” – it is a pure generalized biased speculation and opinion that needs to be better explained or justified.
H. References
Overall – The majority of the references need to be verified as follows:
- References 15 and 34 are the same; same for references 7 and 11; 16 and 35; 18and 44; 19 and 54; 39 and 50; 38a (I have named it with a because they did not have any number) and 49; 70 and 75; 71 and 76; 72 and 77.
- Reference missing: # 20, 73;
- References without a number therefore I have assigned a small a: 38a and 41a;
- Reference incomplete: # 41.
- In the first 3 paragraphs of the discussion section the references #20, 21, 21, 22, 27 and 28 are either incomplete, inverted and do not correspond with the description.
Other suggestions to further improve the significance of this study
- Because of the length of the study (6) years, it would have been important to see the overall growth and the general neurologic outcome of twins breastfed for the longest time compared with twins that did not benefit of such nutrition. There is a mention in the discussion but it is not clear what the final message is: “ Even more marked was the statistically significant correlation between birth weight and the duration of feeding with both total and exclusive breast milk with higher percentages in the category of premature babies weighing <1735 grams.” – was higher incidence of breastfeeding correlated also with higher weight gain?
- There is an interesting comment in your discussion: “In the last year, aid has failed due to forced isolation for the pandemic and this underlines the precious role of grandparents and the importance of this age category so tormented by the COVID 19 pandemic.” – it would be interesting to mention that it is one year out of 6 of observation, it has been an exceptional time and hopefully limited in time, but you should make a point on connectivity and family network in relation to the success of longer breast feeding. Because you state that half had help for a limited period and other half did not receive help from other family component, could you make a comparison between the previous 5 years of observation with the Covid-year of observation? State facts and interpretation of your observation. Also since you cite examples of other studies, could you compare or interpret your own data?
- You have mentioned that: “A protocol validated by psychologists for stress reduction and exaltation of what is positive in the family welcome of twins and their breastfeeding should be standardized.” – What did you see in your cohort of mothers? Mother’s mental health could have also been evaluated during the process: and see if there exist a difference in longer term mental health of motivated mothers (as you described in your study), versus non-motivated mothers.
- At the end of the discussion you mention: “We know that pregnancy and birth of twins are already in themselves conditions of risk; it is our job not to add other risks such as those related to stress and non-breastfeeding.” – How did you do it, could you explain better about what your ‘training’ consists of?
- Additionally you say that: “In actual fact, this territory is totally lacking in professional figures dedicated to breastfeeding other than psychological and material support of women who have just given birth.” – You say you don’t have a multidisciplinary team following these mothers, what the psychological support do for these mothers? It is a good basic support, do all the mothers have access to it? Maybe all you need would be an additional figure, like a nurse lactation consultant that can support at home the women with breastfeeding and understanding of the reading material? What else could you implement as your conclusion and suggestion to others that share the same health care structure? You also say that women that have strong personal believes have breastfed for more than 12 months, so do you propose for the future, when a mother is pregnant with twins (or multiples) a screening to select whether a mother need a low, moderate or higher level of support?
Author Response
x
Dear Dr. Quitadamo,
Your manuscript has been reviewed by experts in the field and we request that you make major revisions before it is processed further. Please revise your manuscript according to the reviewers' comments and upload the revised file within 10 days. Please click on the "Peer Review Reports" below to find the reviewers' comments and the version of your manuscript to be used for your revisions.
Review of paper entitled: "Twins breastfeeding and use of human milk. Factors associated with duration".
This is a mixed quantitative and qualitative analysis of 127 newborns and their mothers recruited over a span of 6 years (inclusive of the last Covid -related year), and where their trend of breastfeeding duration was assessed in correlation to maternal, perinatal and neonatal influencing factors.
It is an interesting study that covers the description of a local population within one of the regions of Italy and gives a narrative slice of the lack of health care organization in the region of Puglia (southern Italy). Such study could be analyzed and compared by colleagues in similar regions and provinces.
Nevertheless, in order to allow colleagues and other researcher to fully benefit of this analysis, and ensure reproducibility, I would request a complete rework as follows point by point.
Title – Conveys unclear meaning. The use of breastfeeding and use of human milk in the same phrase is redundant if not confusing. Suggest revising title and make it a single phrase. I would also include in the title the fact that is a local, single experience within one of the Italian regions.
The title, abstract and conclusions were rewritten. The discussion has been radically changed. The other parts of the manuscript have undergone significant changes
- Abstract – needs to be majorly reworded and clarified
The abstract has been completely rewritten according to the indications
Background: Over the past year, there has been a rise in twin births. The WHO and the others recommended breast feed milk for all newborns for at least 6 month. They stated that it is pos-sible to meet the nutritional needs of two or more newborns with only one mother's milk . The quality of the evidence available from multiple studies has been inconclusive and therefore led to controversial interpretations and practices. More information would be desirable about the factors that influence or lead to the initiation and interruption of breastfeeding. Study: The aim of this study is to evaluate the incidence of twin births, to analyze the feeding of multiples with breast milk and to evaluate the correlation between maternal, perinatal and neonatal variables with breastfeeding and its duration. The study was performed in the Neonatal Intensive Care Unit, Casa Sollievo della Sofferenza Foundation, San Giovanni Rotondo. 61 women have been enrolled who have had a twin birth between 2015 to 2020 with a newborn sample of 127. New-born data were retrospectively collected by informatic database and breastfeeding information were collected by a questionnaire. Results: the percentage of twins out of the total number of births over the years has almost doubled from 1.28% in 2015 to 2.48% in 2020 and the 88% are premature. Infants of lower gestational age and weight, born to multiparous, more mature and medium-high schooling mothers received breast milk for a longer period. 18.1% received breast milk for more than 6 months and 6.3% received it for more than 12 months. 35% of women explained that the interruption of breastfeeding was due to the insufficient milk pro-duction and 41% to the stress and difficulties in managing the twins. In the qualitative part of the study, the narration of the mothers of the twins reveals, for many of them, the awareness of the importance of breastfeeding and the efforts made to try to give breast milk, but also fears about the quantity of milk and satiety of their children. Conclusions: it is important to identify the factors both favoring and obstructing maternal milk feeding and the activation of a network of training and support for mothers after discharge, with particular regard to the categories found to be less inclined.
- Introduction
- “since multiple pregnancy and birth increased the risk of adverse perinatal and postnatal outcomes” – please provide a reference.
These references have been added
Larroque et al.,2004; Delobel-Ayoub et al., 2009; Monden and Smits,2017
- “The most widespread knowledge is that one mother's milk is not enough for two babies even though the WHO and other authorities state that it is sufficient for feeding multiple babies [19]. “ – could be better phrased to lessen bias. Maybe this is very strong peculiar belief in their Italian region? The authors must explain it up front, especially if it is the experience that they have noticed in their practice. The reader needs to understand where this ‘knowledge’ comes from! Who has the ‘most knowledge’? Maybe is not knowledge, maybe the authors mean a myth? Please clarify accordingly.
The sentence has been changed .
“In our experience the myth that one mother's milk is not enough for two babies is still present, even though the WHO and other authorities state that it is sufficient for feeding multiple babies”.
- “Few and conflicting studies have evaluated the breastfeeding rates and factors related to weaning in twins. Our results appear in some ways encouraging, in others disappointing, but they represent a very important source of suggestions, with the reporting of particular cases worthy of being known.” – this whole paragraph should be in the conclusion section of the manuscript.
This paragraph has been moved to the conclusions with some changes
- Methods – I would divide the methods as follows in a very structured way: a) Study design and Settings; b) Participants; c) Ethics and data collection; d) Data analysis.
The majority of the data gathered information could be easily assembled into a table format (i.e. newborns and maternal demographics, exclusion criteria), with the aid of an algorithm you can describe about the 127 subjects, the total numbers of recruited babies at the beginning, the one lost to recruitment (for exclusion criteria or other reasons and so on), the one lost to follow-up, the one non-responding to questionnaire etc. Try to explain everything narratively is too long and confusing for the reader. A comprehensive and well organized table will give justice to the study, and would be easier for the authors to describe it.
We have set the paragraph of methods according to the indications.
We have added an algorithm that specifies the recruiting stages.
2.1. Study Design –
- The authors switch from“127 individuals”, then describe “127 twins”, then “the mothers of 127 twins were contacted by telephone…”: so who are the subjects of your study? I would say both the newborn for quantitative analysis and the mothers that are 63.5(?) adult subjects for your qualitative/quantitative study. You need better explain the number of the pediatric and adult subjects participating to your research.
We have improved the description of the data, in particular of the results by specifying that the research subjects are represented by 127 twin babies and 61 mothers
- In the first paragraph the authors explain the study as a “direct contact with the twins’ mothers” – What does it means?
Direct contact with the mothers took place mainly orally by telephone but the answers to the questionnaire were given by email. This aspect has been specified and added in the text.
- Please immediately explain that it was a questionnaire, as I see it later on the 2 paragraph another related piece of information: “…qualitative data obtained through longitudinal interviews using a semi-structured questionnaire and the extrapolation of socio demographic, biomedical data….three specific database”, lastly I found again the study method in the Data collection section as follows: “The interview to the mothers of the twins was held on the phone and the questionnaire contained 10 questions.” – It is way too scattered and it would be better to follow my suggestion on how to present your data. Were the interviews recorded and transcribed? Who reviewed the questionnaire to be sure that was easy to understand etc?
Each woman was sent a questionnaire by email after telephone contact. In particular, the mothers of the twins were contacted by telephone for the proposal, the oral consent to the participation of the study and the indications on the completion of the questionnaire. In some cases, more telephone conversations followed for clarification on the merits of the questions in the questionnaire. Feedback on understanding the questions was still positive
- The whole paragraph: “The main focus was on the breastfeeding of twins…in order to assess the changes of recent years” should go either in a section called aim of the study or in the discussion, as opinions and experiences can enrich that part of the manuscript.
This paragraph has been deleted and the scope of the study has been changed.
- When the author talk about the ‘three specific database” they are meaning local, regional, national or else? Not clear please specify.
It was specified that the three databases relate to a fairly widespread computer system throughout the country, one is a system in use by the regional administration and a dedicated database for tracking milk bank data.
“from three specific database of national, regional, internal use (Neocare –NICU, SISWEB-regional informatic system, HMB database)”.
- The semi-structured questionnaire (Tab 2) is not provided for reviewer’s analysis in the pdf article; and it is rather described by the authors, as a survey with a yes or no answers. The description of a longitudinal phone call with the questionnaire does not explain the true content of the interview (are they asking open questions during their interviews?) and/or an interview format in which either transcripts of comments from the targeted population is then judged for credibility. How did the researcher reach a thematic consensus? The examples of verbatim that is possibly part of the qualitative interview with the mothers, should be better introduced, interpreted to the main goal of this study, and more examples would be probably appreciated by the readers.
The term semi-structured was not appropriate and was deleted. We added the questionnaire. We have also improved the description of the qualitative part of the study by adding more literary examples.
2.2. Setting –
- “disease incompatible with lactation” – please mention examples in the exclusion criteria as I have mentioned above, in the overall comment.
Examples have been added
- The word ‘bank milk’ can alternatively be called ‘human donor milk’.
This change has been made
- “For healthy full-term infants, the first breast attack occurs a few minutes”…please change to ‘the first latch on the breast, occurs in few minutes after birth, while doing skin-to-skin with the mother for the first two hours of life…”
This sentence has been deleted
- “Nevertheless, the situation of newborns’ discharge is critical in our region. In actual fact, this territory is totally lacking in professional figures dedicated to breastfeeding other than psychological and material support of women who have just given birth. It is known that the first weeks after childbirth are particularly difficult for mothers in a particular physical and psychological stress and poor hormonal balance where the only help, when available, falls to family members….to twin pregnancies” – The English structure should be reviewed, and the whole paragraph should be moved to the discussion section.
The paragraph has been revised and moved to discussion.
2.3. Study participants Sample –
- The title to this methodology section ‘Sample’ should be replaced by – My suggestion here is that it would be better explained to the readers as a schematic algorithm that includes the total number of recruited individuals (newborns and mothers), the numbers of subjects excluded with exclusion criteria well stated. Who participated to the questionnaire/telephonic interview, how many lost in follow up etc.
Title replacement has been made.
The schematic algorithm of the recruited subjects has been added.
- 6-year period between 2015 and 2020/21.” – does the period end in 2021? Authors should disclose the exact period of time included in their recruitment of subjects with twin pregnancy.
The 6-year period ends on December 31, 2020 and has been specified. 2021 was considered only to extract the data on the incidence of twin births in order to better highlight the trend over the years.
2.4. Definitions –
- One comment on the way to breastfeed: what about twins that were bottle fed? Or half breastfed and ½ bottle-fed? Would other caregivers help out? Other human milk than mother? Could have this change the overall result?
- The word trio has been mentioned in the results of the manuscript – not clear if it is referred to the trio mother & twins or 3 newborns, please explain.
.This paragraph was not functional to the text and it was deleted
The trio refers to triplets.
2.5. Data collection –
- Definitely the authors should provide a picture of the questionnaire with the 10 questions, and then just refer to it within the text in lieu of the narrative. Maybe was contained in referred Table 2. If the questionnaire is already included in Table 2, please ignore this comment.
The narration was deleted and the picture of the questionnaire was provided
2.6. Data analysis –
- only the quantitative data analysis seems described, while in the methods a qualitative research was mentioned. Going back to the data collection and in connection with this point, and because the table 2 describing the questionnaire (?), is not fully available, it seems that questions were not open, but just a yes/no answer, which matches with a survey rather than a recorded narrative interviews? It would be nice to know after the yes/no response, if at the end of each questions a space for free comments?
The paragraph has been rewritten and the method used to obtain information from mothers has been better specified. The questionnaire was made clear by adding the text of the questions.
- Results – In the pdf copy of the manuscript the 5 tables summarizing the data results were not included. The narrative attempt of describing the data is confusing, redundant and difficult to follow because of lack of statistical comparisons when looking at the results. It ends up in a list of numbers and percentages that dilutes in details the interpretation of it all.
- This section is divided in 3 parts: a) population: very confusing when describing twins babies instead of number of ‘sets of twins’. – Please stick with one term, and eventually add it in your Definitions section; b) Use and duration of MM; c) Reasons for not breast feeding and availability of help: the last one it is a very skimpy section. This is surprising because it reflects the qualitative part of the whole study, please expand.
We have tried to improve the results paragraph by schematize them and make them more understandable
The qualitative part has been expanded with the addition of several stories.
- There are sentences that mix babies with mothers demographic and are difficult to extrapolate. Example: “55 newborns have mothers with high school diplomas (49.5%), 33 (29.7%) with university degrees…” and “The mothers of 53 newborns are housewives, 25 are employed, 16 are self-employed, 19 workers and 8 are unemployed.” And again “The mothers of 98 newborns are Italian (94.2%) and 6 (5.8%) come from Romania”. – It would be better to explain it as follows: “In our cohort of mothers with twins, there were 49.5% that obtained the high school diploma…” and so on.
Changes to improve the text according to the indications have been made.
- Authors emphasize in the title and intro the importance of breast feeding as it pertains to the twin cohort, yet, in the description of their population results there are described triplets, and quadruplets, namely “ The twin births are 114 (89.8%), the triplets 9 (7.1%), and the quadruplets 4 (3.1%).” ???
We have better specified that in the context of twin pregnancies: 114 (89.8%) were born from twin births, 9 from triplets and 4 from quadruplets.
- “The category of premature babies” – Is the title of another section.? Because the GA has already been described in the results section, this part gives a confusing way and version of looking at the whole percentage of prematurity among twins, and should be eliminated.
In this section, the feeding data by gestational age and birth weight have been detailed because we consider it a crucial aspect and we have not described it before. We tried to lighten it.
- In the section Use and duration of MM the following sentence is incomprehensible and therefore of difficult interpretation “The couple and the 27-week of GA trio received breast milk for 36 months and 6 months, respectively, and were breastfed. Both mothers donated milk to the bank, the first up to the maximum donation term which is 6 months and donated more than 50 liters of milk and the mother of the 3 twins donated about 5 liters during the hospitalization.” – please rewrite in comprehensible English.
This paragraph has been rewritten
- The authors write “In the statistical comparison (tables education, birth weight 1,2, gestational age 1,2,…) between the different variables…” – The phrase never ends and there is an entire 10-lines paragraph that doesn’t have a real meaning. Needs to be done again.
The sentence has been made schematic
- Much of the results covered in the Discussion section could be moved back to Results section.
They are the same results described in the results section while they have been simplified in the discussion paragraph.
- Discussion– Summarizing, the discussion section that appears overly detailed and scattered over many mini-topics, not focused on the title (factors associated with duration) and conclusions are not sufficiently balanced, with the ultimate goal which again is vague in accordance to your initial aims. In the purpose of the study the authors wish to explore trends of breastfeeding in women with twins; therefore, to be consistent, it would be important to summarize in the conclusion if an upward trend was observed, and which are the novel factors (if any) that influences the duration of breastfeeding of twins. Overall the whole Discussion section needs to be reviewed by a professional that is comfortable with the English grammatical structure and composition. Many sentences are awkward and do not convey well their meaning. Discussion lacks structure for better readability.
- The overall discussion has an exceptional length, the narrative part is too long and full of details that are diluting the main message of the authors.
- Improve the use of verbatim by following a logical sequence.
- Do not mix arguments: i.e. the incidence of twin pregnancies should be separated from the incidence of breastfeeding.
- After every mini review of the literature, there is one brief sentence about the local study, but at one point it needs to be paraphrased and integrated into the discussion text, with particular attention to a ‘continuity’ to help the reader understand what is relevant. I would suggest 3 possible actions to be considered and chosen to improve the reading:
- Please summarize the different paragraphs into one general sentence: the first two paragraphs speak about increased incidence of multiple pregnancy with too many details that can be confusing; the authors should explain that the increased rate of multiple pregnancies has been trending up in the last #of years, you can name the different references for the different countries involved in similar epidemiologic studies (UK, U.S., Italy, Japan etc), then focus on your local data to compare.
- Maybe you should structure the discussion in subtitles:Incidence of multiple births; Incidence of Multiple births and breastfeeding; Prevalence and associated factors of breastfeeding twins and so on. The authors could put in order their review of the literature and then make a comparison with what they have found. It may be easier to follow.
The paragraph in the discussion has been toned down following the suggestions. Subchapters have been created
- Review of the literature and interpretation of the data section. Once restructured, and summarized with comparison with local data for every topic, this Discussion will turn into a review. If the authors choose to do so, they need to remember to put the word ‘review’ in the title. The discussion will then be only focused on answering in regard of the aim of their study and on novel observations found in their local cohort.
“…only a few studies have focused on breastfeeding twins and, consequently few evidence based interventions have been identified that are specific to mothers of twins and higher order multiples and of preterm twins. Not only, these few studies have also produced controversial results.”; I don’t agree with the author statement that there are few studies on breastfeeding multiples. In this authors’ reference list I found at least 18 articles; it also exist national and international guidelines, written by different Pediatric Societies available on line, a Cochrane review. Just reviewing how different countries have tried to support education on breastfeeding twins, I found 10 RCT supporting educations. Lastly, published analyses about factors that benefit or delay or interrupts breastfeeding in twins it is not scarce. Therefore I would suggest the authors to rephrase the paragraph with something like: “The quality of the evidence available from multiple studies has been inconclusive and therefore led to controversial interpretations and practices.”
The proposed change to this paragraph has been made..
- For your own information, please look at Multiple Births Foundation 2011; McAndrew 2012; Bonet 2011; Yokohama, 2006, Nursan Cinar, J Health Popul Nutr. 2013; Nursan Cinar, Iran J Pediatr. 2016; just to mention a few qualitative studies on the same subject.
The authors write: “In our experience [42], if motivated and initiated early on breast stimulation, even mothers of extremely premature infants can exclusively feed their infants…” and then again “Moreover multiparous women, probably because of their better organizational skills…” – similar statements found in the whole discussion section. The reader could wonder if these statements are the reflection of some other observations in which the authors have studied and/or compared within their studied population different modes of interventions, but nowhere in the manuscript they justify these opinions/statements appropriately. Did you consider also as a ‘high motivation’ the in vitro fertilization factor?
The first is an observation based on experience that was described in another of our studies of which there is a reference.
The second observation refers to another of our studies where we described that in our experience as a milk bank, multiparous women donated longer. The expression that could give the impression of expressing a judgment has been eliminated.
- Please replace the bolded word: “Other papers [55] have found no breastfeeding differences between singleton and twin preterm infants.” with ”Other authors [55] have found…”
This replacement was made..
- Results like the ones I have listed below (just few examples but there are many more), belong to the results section of the manuscript rather than the discussion section; use the discussion to compare and interpret your data.
- “In this study, with a p value of 0.015, the correlation between the variable parity and the months of breastfeeding was statistically significant, while this significance was not confirmed with reference to exclusive breastfeeding”;
- “The mean age of the mothers of a cross-sectional survey involving 185 mother-twin pairs [22] was 30.18+/-1.29 years. In our population the average is higher, 35 years with the median of 36.”;
- “Statistically significant was the association between the two categories of maternal age (more or less than 36 years) both for the duration of feeding with breast milk and with exclusive breastfeeding.”
“There was a statistical positivity (p-value 0.028) in the correlation of gestational age with the exclusive feeding with breast milk with a higher incidence among preterm gestational age <33 weeks.”
We have indicated these results to elaborate them for discussion also with reference to the data in the coating process. But we have summarized them.
- “In our population we must give importance to the case of the foreign woman with a strong motivation towards breastfeeding and the need to feed with her milk… subsequently complemented thanks to her conviction and determination, is also noteworthy.” – these paragraphs are commendable, but would be more suited to acknowledgements section.
- “Since mothers of multiples cannot or do not want to breastfeed for a long duration may result in a significant missed opportunity, particularly as the number of multiple births continues to grow.”: this statement alone in the context is pretty obvious, and should be completed by the authors with another sentence that gives meaning to this ‘discovery’ (how do you want to support these mothers through these changes?) or better be deleted.
Authors should ask themselves the following questions and structure their discussion upon: What do the results mean for the problem and hypothesis of our study? What are the implications (if any) for the targeted population?
The discussion was largely changed. References to the literature have been heavily scaled down to be eventually used in a future review. We removed the comments that appeared to be opinions. We have deepened the interpretation of the data.
- Conclusion– Major rework is needed.
- The first sentence should be broken up to smaller sentences and grammatically edited for readability.
- The authors tried to formulate a conclusion: “The ultimate goal is to identify and share a protocol of information and support, especially for women who fall into those categories less accustomed to this practice…” – The authors should try to answer to questions like: “How do our findings add to the current literature?” or again “Any suggestions for future research?”
- “For this purpose, given the current lack of data, mostly linked to the widespread belief but denied by the WHO, of the impossibility of meeting the nutritional needs of two or more newborns with exclusive mother's milk.” – This is already appearing in the abstract (please see my comments in the abstract section above), and should be rephrased in the proper English form.
The conclusion has been totally rewritten according to the indications.
x
Conclusion
The incidence of twin births has doubled in 6 years, in line with Italian and European data. The use of breast milk and the breastfeeding of twins recorded percentages below the standards indicated by the WHO. The maternal factors that were statistically significant correlated with the duration of breastfeeding were age, the educational level, the parity with better results obtained with older mothers, with a higher educational level and multiple parity. These data suggest the desirability of greater support for mothers who are younger and with a lower cultural level. Multiparity, on the other hand, does not represent a prejudice to prolonged breastfeeding.
The neonatal factors that impacted breastfeeding rates were birth weight and gestational age with the best performance observed for the lower weight and lower gestational age categories of newborns. This confirms the effectiveness of standardized protocols for the promotion of milk production by mothers of premature babies. Care dedicated to breastfeeding should be intensified for mothers of siblings of greater weight and gestational age.
The main reasons for interrupting breastfeeding are the insufficient milk production and stress and difficulties in managing the twins. The concept that breast milk may be sufficient to breastfeed twins should be more widespread, as reiterated by the WHO.
About 50% of mothers of twins did not receive any help. The presence of expert people for the support of mothers is not even named, confirming a welfare gap in the area. Support was provided within the family and the most important role was played by grandparents, traditionally fundamental social figures and particularly in this historical period in our area. Family support had a statistically significant impact on the duration of breastfeeding.
In the qualitative part of the study, the narration of the mothers of the twins reveals, for many of them, the awareness of the importance of breastfeeding and the efforts made to try to give breast milk, but also fears about the quantity of milk and satiety of their children.
It would be interesting to evaluate the outcome of twins fed with breast milk and those with formulated milk and this will be the subject of a future study.
cx
- Limits and opportunities– This whole session should go before the conclusion.
- The phrase “…which remains disappointing all over the world” – it is a pure generalized biased speculation and opinion that needs to be better explained or justified.
The sentence has been deleted.
- References
Overall – The majority of the references need to be verified as follows:
- References 15 and 34 are the same; same for references 7 and 11; 16 and 35; 18and 44; 19 and 54; 39 and 50; 38a (I have named it with a because they did not have any number) and 49; 70 and 75; 71 and 76; 72 and 77.
- Reference missing: # 20, 73;
- References without a number therefore I have assigned a small a: 38a and 41a;
- Reference incomplete: # 41.
- In the first 3 paragraphs of the discussion section the references #20, 21, 21, 22, 27 and 28 are either incomplete, inverted and do not correspond with the description.
The corrections to the references remaining after the text modification have been made. They have been double-checked to avoid errors and repetitions.
Other suggestions to further improve the significance of this study
- Because of the length of the study (6) years, it would have been important to see the overall growth and the general neurologic outcome of twins breastfed for the longest time compared with twins that did not benefit of such nutrition. There is a mention in the discussion but it is not clear what the final message is: “ Even more marked was the statistically significant correlation between birth weight and the duration of feeding with both total and exclusive breast milk with higher percentages in the category of premature babies weighing <1735 grams.” – was higher incidence of breastfeeding correlated also with higher weight gain?
- There is an interesting comment in your discussion: “In the last year, aid has failed due to forced isolation for the pandemic and this underlines the precious role of grandparents and the importance of this age category so tormented by the COVID 19 pandemic.” – it would be interesting to mention that it is one year out of 6 of observation, it has been an exceptional time and hopefully limited in time, but you should make a point on connectivity and family network in relation to the success of longer breast feeding. Because you state that half had help for a limited period and other half did not receive help from other family component, could you make a comparison between the previous 5 years of observation with the Covid-year of observation? State facts and interpretation of your observation. Also since you cite examples of other studies, could you compare or interpret your own data?
The clarification has been inserted
The concept of the exceptionality of the year 2020 linked to the pandemic has been added.
The comparison has been added.
- You have mentioned that: “A protocol validated by psychologists for stress reduction and exaltation of what is positive in the family welcome of twins and their breastfeeding should be standardized.” – What did you see in your cohort of mothers? Mother’s mental health could have also been evaluated during the process: and see if there exist a difference in longer term mental health of motivated mothers (as you described in your study), versus non-motivated mothers.
It is an interesting idea that we reserve the right to deepen in the future, because we believe that the assessment of the mental state requires specific skills and more time with additional questions or dedicated tests
The correlation between the presence of help and the duration of breastfeeding has been added.
The availability of help in the family context was correlated with a significant lengthening of the duration of breastfeeding. This aspect has been deepened and the results have been reported, reference has been made in discussion and in the conclusion
At the end of the discussion you mention: “We know that pregnancy and birth of twins are already in themselves conditions of risk; it is our job not to add other risks such as those related to stress and non-breastfeeding.” – How did you do it, could you explain better about what your ‘training’ consists of?
The sentence has been deleted.
Additionally you say that: “In actual fact, this territory is totally lacking in professional figures dedicated to breastfeeding other than psychological and material support of women who have just given birth.” – You say you don’t have a multidisciplinary team following these mothers, what the psychological support do for these mothers? It is a good basic support, do all the mothers have access to it? Maybe all you need would be an additional figure, like a nurse lactation consultant that can support at home the women with breastfeeding and understanding of the reading material? What else could you implement as your conclusion and suggestion to others that share the same health care structure? You also say that women that have strong personal believes have breastfed for more than 12 months, so do you propose for the future, when a mother is pregnant with twins (or multiples) a screening to select whether a mother need a low, moderate or higher level of support?
We don't have dedicated psychological support. We explained ourselves incorrectly. We followed the suggestions and proposed screening based on maternal motivation.

Reviewer 2 Report
Review of Twins breastfeeding and use of human milk. Factors associated with duration.
Title – grammatically incorrect and conveys an unclear meaning. Suggest revising title.
Abstract:
“The possibility of being fed with breast milk is completely insufficient compared …” – not in all cases, a severe statement. In science one tries to avoid absolute terms and categorical opinions.
“Study: Totally…” – should be total.
“…medium-high schooling mothers…” – meaning not clear.
Abbreviations “EMM” and “MM” – not clarified.
“to counter the widespread mistaken belief, denied by the WHO, that it is impossible to meet the nutritional needs of two or more newborns only with mother's milk” – could be better phrased to lessen bias.
Introduction:
“The most widespread knowledge is that one mother's milk is not enough for two babies even though the WHO and other authorities state that it is sufficient for feeding multiple babies [19]. “ – could be better phrased to lessen bias.
Methods:
2.2. Setting – “In actual fact, this territory is totally lacking in professional figures dedicated to breastfeeding other than psychological and material support of women who have just given birth. It is known that the first weeks after childbirth are particularly difficult for mothers in a particular physical and psychological stress and poor hormonal balance where the only help, when available, falls to family members.” – no reference provided for the statements. And once again, a severe statement. In science there is no truth, just data that approximates the underlying natural principles at play. Plus, redundant use of English. If anything is fact, then it is actual. “Totally lacking” is non-scientific since the inventory of data being referred to is not stipulated. Because some data is not apparent it does not follow the data do not exist.
2.3. Sample – “6-year period between 2015 and 2020/21.” – does the sample period end in 2021? If so, if authors want to include 2020, they should also include every year from 2015 onwards.
Results:
The reviewer copy of the manuscript did not include Tables.
The summary of results is not detailed enough. For example, in the Results section, the authors write “the statistical comparison was made between these two age groups.” Please describe the statistical comparisons and the results derived from that. Additional descriptions would also be helpful, where abbreviations are used. Much of the results covered in the Discussion section could be moved to Results section.
The results section is awkwardly formatted, e.g. “The category of premature babies” – is this a title?
Discussion:
Authors emphasize the importance of breast feeding as it pertains to the twin cohort. However, the use of study quotes could be improved upon, if only to be added to supplementary material, or paraphrased and integrated into the discussion text.
“Moreover multiparous women, probably because of their better organizational skills” –statements similar to that shown in bold should be avoided. Judgemental, how were these skills identified and assessed as better or worse? Cite the relevant reports, otherwise remove from manuscript if you want the latter to appear more scientific.
“In our population we must give importance to the case of the foreign woman with a strong motivation towards breastfeeding and the need to feed with her milk… subsequently complemented thanks to her conviction and determination, is also noteworthy.” – while admirable, these paragraphs would be more suited to acknowledgements.
Conclusion:
The first sentence should be broken up to smaller sentences and grammatically edited for readability.
“For this purpose, given the current lack of data, mostly linked to the widespread belief but denied by the WHO, of the impossibility of meeting the nutritional needs of two or more newborns with exclusive mother's milk.” – repeat from abstract. The manuscript requires rigorous rewriting to make it cohesive, clear and convincing.
Limits and opportunities:
“…which remains disappointing all over the world” – could be stated better to reduce bias. Again, the word ‘all’ is a give-away, not a scientific statement. Have the authors been to Axel Heiberg Island to test whether their statement applies t that region as well?
Overall:
Significant English revisions are needed. Throughout the manuscript, authors state their opinions instead of unbiased observations of results in reference to peer-reviewed publications in high impact factor journals.

Author Response
Reviewer 2
Review of Twins breastfeeding and use of human milk. Factors associated with duration.
Title – grammatically incorrect and conveys an unclear meaning. Suggest revising title.
Abstract:
“The possibility of being fed with breast milk is completely insufficient compared …” – not in all cases, a severe statement. In science one tries to avoid absolute terms and categorical opinions.
“Study: Totally…” – should be total.
“…medium-high schooling mothers…” – meaning not clear.
Abbreviations “EMM” and “MM” – not clarified.
“to counter the widespread mistaken belief, denied by the WHO, that it is impossible to meet the nutritional needs of two or more newborns only with mother's milk” – could be better phrased to lessen bias.
The abstract has been totally rewritten
Background: Over the past year, there has been a rise in twin births. The WHO and the others recommended breast feed milk for all newborns for at least 6 month. They stated that it is pos-sible to meet the nutritional needs of two or more newborns with only one mother's milk . The quality of the evidence available from multiple studies has been inconclusive and therefore led to controversial interpretations and practices. More information would be desirable about the factors that influence or lead to the initiation and interruption of breastfeeding. Study: The aim of this study is to evaluate the incidence of twin births, to analyze the feeding of multiples with breast milk and to evaluate the correlation between maternal, perinatal and neonatal variables with breastfeeding and its duration. The study was performed in the Neonatal Intensive Care Unit, Casa Sollievo della Sofferenza Foundation, San Giovanni Rotondo. 61 women have been enrolled who have had a twin birth between 2015 to 2020 with a newborn sample of 127. New-born data were retrospectively collected by informatic database and breastfeeding information were collected by a questionnaire. Results: the percentage of twins out of the total number of births over the years has almost doubled from 1.28% in 2015 to 2.48% in 2020 and the 88% are premature. Infants of lower gestational age and weight, born to multiparous, more mature and medium-high schooling mothers received breast milk for a longer period. 18.1% received breast milk for more than 6 months and 6.3% received it for more than 12 months. 35% of women explained that the interruption of breastfeeding was due to the insufficient milk pro-duction and 41% to the stress and difficulties in managing the twins. In the qualitative part of the study, the narration of the mothers of the twins reveals, for many of them, the awareness of the importance of breastfeeding and the efforts made to try to give breast milk, but also fears about the quantity of milk and satiety of their children. Conclusions: it is important to identify the factors both favoring and obstructing maternal milk feeding and the activation of a network of training and support for mothers after discharge, with particular regard to the categories found to be less inclined.
Introduction:
“The most widespread knowledge is that one mother's milk is not enough for two babies even though the WHO and other authorities state that it is sufficient for feeding multiple babies [19]. “ – could be better phrased to lessen bias.
The sentence has been changed .
“In our experience the myth that one mother's milk is not enough for two babies is still present, even though the WHO and other authorities state that it is sufficient for feeding multiple babies”.
Methods:
2.2. Setting – “In actual fact, this territory is totally lacking in professional figures dedicated to breastfeeding other than psychological and material support of women who have just given birth. It is known that the first weeks after childbirth are particularly difficult for mothers in a particular physical and psychological stress and poor hormonal balance where the only help, when available, falls to family members.” – no reference provided for the statements. And once again, a severe statement. In science there is no truth, just data that approximates the underlying natural principles at play. Plus, redundant use of English. If anything is fact, then it is actual. “Totally lacking” is non-scientific since the inventory of data being referred to is not stipulated. Because some data is not apparent it does not follow the data do not exist.
We have eliminated all phrases that could be interpreted as opinions
2.3. Sample – “6-year period between 2015 and 2020/21.” – does the sample period end in 2021? If so, if authors want to include 2020, they should also include every year from 2015 onwards.
The data has been specified. The period covered by the study includes 6 years, from 2015 to 2020.
Results:
The reviewer copy of the manuscript did not include Tables.
The summary of results is not detailed enough. For example, in the Results section, the authors write “the statistical comparison was made between these two age groups.” Please describe the statistical comparisons and the results derived from that. Additional descriptions would also be helpful, where abbreviations are used. Much of the results covered in the Discussion section could be moved to Results section.
We have detailed and schematized the results
The results section is awkwardly formatted, e.g. “The category of premature babies” – is this a title?
We have reformatted, to make it easier to understand the results
Discussion:
Authors emphasize the importance of breast feeding as it pertains to the twin cohort. However, the use of study quotes could be improved upon, if only to be added to supplementary material, or paraphrased and integrated into the discussion text.
The discussion was largely changed. We removed the comments that appeared to be opinions. We have deepened the interpretation of the data.
“Moreover multiparous women, probably because of their better organizational skills” –statements similar to that shown in bold should be avoided. Judgemental, how were these skills identified and assessed as better or worse? Cite the relevant reports, otherwise remove from manuscript if you want the latter to appear more scientific.
It is a fact that comes from our experience but we still preferred to delete the sentence to follow the advice.
“In our population we must give importance to the case of the foreign woman with a strong motivation towards breastfeeding and the need to feed with her milk… subsequently complemented thanks to her conviction and determination, is also noteworthy.” – while admirable, these paragraphs would be more suited to acknowledgements.
Since the study includes a qualitative part, we decided to leave the short story after having improved it.
Conclusion:
The first sentence should be broken up to smaller sentences and grammatically edited for readability.
“For this purpose, given the current lack of data, mostly linked to the widespread belief but denied by the WHO, of the impossibility of meeting the nutritional needs of two or more newborns with exclusive mother's milk.” – repeat from abstract. The manuscript requires rigorous rewriting to make it cohesive, clear and convincing.
The conclusion was totally rewritten indicating the significant results of the study.
Limits and opportunities:
“…which remains disappointing all over the world” – could be stated better to reduce bias. Again, the word ‘all’ is a give-away, not a scientific statement. Have the authors been to Axel Heiberg Island to test whether their statement applies t that region as well?
The sentence has been deleted
Overall:
Significant English revisions are needed. Throughout the manuscript, authors state their opinions instead of unbiased observations of results in reference to peer-reviewed publications in high impact factor journals.
The manuscript has been heavily modified in all its parts both in terms of content and English form.

Round 2
Reviewer 1 Report
This is the second Review of Feeding the twins with breast milk. Quantitative and qualitative study in a NICU in Southern Italy with the milk bank.
Authors – Please replace “medicine student” with "Medical student, Faculty of Medicine, University of (please complete the full affiliation)"
Title – It would be better to revise the title to make it one sentence only. Authors can use “:” to connect sentences.
Here's my suggestion for a simpler title: “Feeding twins with human milk and factors associated with its duration: a qualitative and quantitative study in Southern Italy”- the fact that the milk was obtained from a bank can be specified later in the manuscript, it's not necessary to include in the title.
Abstract – Please bold the words Background, Methods, Results and so on.
“The WHO and the others” is too ambiguous. Perhaps something like “The current scientific consensus is.”
No need to have “Study:” in the abstract. Rather include a “Methods:” section.
The aim is not clear and is filled with too many goals:
- To evaluate the incidence of twin births (for who, where -region, hospital, province - and compared to what -national, European, other Italian regions? Is it your first aim?)
- Analyze the feeding of multiple with breast milk (too generic, analyze how? By looking at the quantity of feeding and or the quality of feedings? i.e. breast milk versus donor milk, or breast milk vs donor milk and duration?)
- To evaluate the correlation between maternal, perinatal and neonatal variables with breastfeeding and its duration. The aim should be rephrased in a simpler fashion, like: “The aim of this study was to evaluate the correlation between maternal, perinatal and neonatal variables with breastfeeding and its duration.”
Instead of “The study was performed in the NICU, casa Sollievo….etc” replace by something simpler, adapted to an abstract like “The study was conducted between 2015 and 2020, in a NICU in Southern Italy (San Giovanni Rotondo, Foggia). Sixty-one women who have given birth to twins were enrolled into the study.”
Please start the Results with: “In our region…” to focus on your local findings for better understanding.
Please consolidate grammar throughout abstract, e.g., after ‘:’ should have first letter capitalized throughout.
Additionally authors should decide about constructing sentences either in the present or past tense, and be consistent throughout the Abstract.
Introduction – Generally poorly written in English even if I am understanding the general intention of the authors. Some paragraphs need to be summarized into simpler and straightforward sentences.
Page 2 – “The latter data…NICU” – no reference for sentence.
“In our experience the myth…” – please rephrase sentence to refer to interpretation of your data only.
Heading “The aim of the study” is unnecessary.
Methods -
2.1. Study Design – “thanks to the data” – could be rephrased.
2.3. Study Participants – Please revise paragraph to reduce the minor grammatical errors, e.g. instead of “to preliminary information” it should be “for”.
What is “Figure_algorithm”? If Figure is available (not visible to the reviewer), please title it as Figure 1.
Authors state that 61 mothers of twins completed the questionnaire, indicating that the total was 127 twins enrolled in the study. If 61 mothers had twins, wouldn’t the total number of twins be 122? Please clarify.
2.5 Ethics and Data Collection – Please provide ethics approval number. Again, “Figure_questionnaire” not visible to reviewer nor correctly labelled. I would suggest if including, please add this to supplementary materials.
Authors say all data collected was recorded in a dedicated database – is this database public? Does it have a name?
Not necessary to include information about meetings.
Results –
Could remove sentence “The results are summarized in tables 1-5.”. Instead, refer to the tables throughout results text. To notice that table 5 is not visible to reviewer.
3.1. Population - Please separate results description by table. For example, include table that showcases the data that “68 (54.4%) of twin babies are first children…”.
3.2. Breastfeeding and use of MM – please include data tables here as well, if applicable. If not data tables available for these results, please create them.
Both sections 3.1. and 3.2. could be more clearly written to showcase results. Lack of flow in writing.
3.3. – For quotations, please refer to instructions here to improve flow: https://apastyle.apa.org/style-grammar-guidelines/citations/quoting-participants
Sentence, “In their story, about 30%...bottle” – how was this analyzed? Is it derived from results table. Could re-write to ”less than half of the mothers”, and remove “in their story”.
Tables with results - Results tables lack sufficient descriptive titles that would allow reader to view them independently of text. Such tables must benefit of changing into pies or graphs?
Please change to 1.28%. In English we use periods (.) instead of commas (,), for decimal points.
In the table legend “Not Disponible Date” needs to be changed to N/A (which means not applicable, not available or no answer).
Discussion - This section is excessive in length. Many of the authors results interpretations could be moved to results section and combined with current results, including their respective headings. It would improve readability of the manuscript and results.
Conclusion - The first sentence should be broken up to smaller sentences and grammatically edited for readability. Also instead of a generalized affirmation “The incidence of twin births has doubled in 6 years…” should be replaced by “ The incidence in our region of…”.
“The neonatal factors that impacted breastfeeding rates were birth weight and gestational age with the best performance observed for the lower weight and lower gestational age categories of newborns.”- the underlined observation requires a deeper introspection by the authors on other social/cultural/psychological and/or anthropological reasons why this happens, and cannot be attributed solely to a standard protocol (see examples of micro-premies in Brazil, where the NICUs have no incubators available and the babies thrive and survive on skin-to-skin care only).
“The main reasons for interrupting breastfeeding are the insufficient milk production and stress and difficulties in managing the twins.” – do the authors think of possible relationship about a ‘certain’ type of stress in relation to the type of delivery (C/S versus vaginal?), the timing or persistence of the stress? The timing and or the persistence of the support?, generally all these affirmations leave the audience wondering about what possibly could be done to change the outcome, this ‘food for thoughts’ should be delineated by the authors.
Acknowledgments - I suggest the following: “This study is dedicated to a special mother who donated her milk to the premature babies in our hospital and to those mothers contributing in donating their breastmilk to the Milk Bank of the “Bambino Gesù” Hospital in Rome. We thank all individuals who advocate and work to ensure that newborns and infants can have the right to a health opportunity represented by accessibility to human milk.”
Overall
Major English writing revisions are still needed, though authors have greatly improved upon their initial manuscript.
Overabundance of minor grammatical errors are present throughout the manuscript.
Few examples: “formulated milk” should be replaced by “formula milk”;
“Fertilization pregnancy” should be replaced by: “assisted reproductive technology (ART)” and so on.
“In the qualitative part of the study, the narration of the mothers of the twins reveals…” should be replaced by “ Qualitative analysis of maternal narrative revealed…”;
“This triggered regrets and guilt. This led to the abandonment of breastfeeding in the first 2-3 months.” Should be made a continuous phrase like: “ This context triggered regrets and guilt, and led to the abandonment of breastfeeding in the first 2-3 months”.
The sentence “The limit of this study” is too optimistic, I would change to “One of the limits of this study…”. Thanks.
Please MUST change: “the pleasure of attaching them to the breast..”, we do not attach babies on the breast we make them LATCH to the breast! THANK YOU!

Author Response
x
Authors – Please replace “medicine student” with "Medical student, Faculty of Medicine, University of (please complete the full affiliation)"
It has been modified
4Medical student, Faculty of Medicine, University of San Raffaele Vita-Salute, Milano, Italy
Title – It would be better to revise the title to make it one sentence only. Authors can use “:” to connect sentences.
Here's my suggestion for a simpler title: “Feeding twins with human milk and factors associated with its duration: a qualitative and quantitative study in Southern Italy”- the fact that the milk was obtained from a bank can be specified later in the manuscript, it's not necessary to include in the title.
Feeding twins with human milk and factors associated with its duration: a qualitative and quantitative study in Southern Italy
Abstract – Please bold the words Background, Methods, Results and so on.
Headings have been bolded
“The WHO and the others” is too ambiguous. Perhaps something like “The current scientific consensus is.”
The change was made according to the indications
No need to have “Study:” in the abstract. Rather include a “Methods:” section.
The word study has been removed
The aim is not clear and is filled with too many goals:
To evaluate the incidence of twin births (for who, where -region, hospital, province - and compared to what -national, European, other Italian regions? Is it your first aim?)
Analyze the feeding of multiple with breast milk (too generic, analyze how? By looking at the quantity of feeding and or the quality of feedings? i.e. breast milk versus donor milk, or breast milk vs donor milk and duration?)
To evaluate the correlation between maternal, perinatal and neonatal variables with breastfeeding and its duration. The aim should be rephrased in a simpler fashion, like: “The aim of this study was to evaluate the correlation between maternal, perinatal and neonatal variables with breastfeeding and its duration.”
The sentence was rewritten in this way:
Aims: The first aim of this study was to analyze the extent of the feeding of multiples with breast milk in the experience of our clinical unit, in terms of incidence and duration. The second objective was to evaluate the correlation between maternal, perinatal and neonatal variables with breast milk feeding rates and duration.
Instead of “The study was performed in the NICU, casa Sollievo….etc” replace by something simpler, adapted to an abstract like “The study was conducted between 2015 and 2020, in a NICU in Southern Italy (San Giovanni Rotondo, Foggia). Sixty-one women who have given birth to twins were enrolled into the study.”
We have modified it as follows:
The study was conducted between 2015 and 2020, in a neonatology / NICU in Southern Italy (San Giovanni Rotondo, Foggia). Sixty-one women who have given birth to twins were enrolled into the study
Please start the Results with: “In our region…” to focus on your local findings for better understanding.
We made this change: "in our clinical center"
Please consolidate grammar throughout abstract, e.g., after ‘:’ should have first letter capitalized throughout.
This grammar change was made.
Additionally authors should decide about constructing sentences either in the present or past tense, and be consistent throughout the Abstract.
We have built all the sentences in the past tense.
Introduction – Generally poorly written in English even if I am understanding the general intention of the authors. Some paragraphs need to be summarized into simpler and straightforward sentences.
The paragraph has been simplified and has been revised in the English construct.
Page 2 – “The latter data…NICU” – no reference for sentence.
The sentence has been deleted
“In our experience the myth…” – please rephrase sentence to refer to interpretation of your data only.
In our experience the belief that one mother's milk is not enough for two babies is still present, even though the WHO and other authorities state that it is sufficient for feeding multiple babies
Heading “The aim of the study” is unnecessary.
The header has been deleted
Methods -
2.1. Study Design – “thanks to the data” – could be rephrased.
Twins birth data (delivery time, parity, birth weight) are obtained using the data collection systems present at NICU (Neocare –NICU, SISWEB-regional informatic system).
2.3. Study Participants – Please revise paragraph to reduce the minor grammatical errors, e.g. instead of “to preliminary information” it should be “for”.
The change has been made
What is “Figure_algorithm”? If Figure is available (not visible to the reviewer), please title it as Figure 1.
The caption Figure 1 has been inserted.
Authors state that 61 mothers of twins completed the questionnaire, indicating that the total was 127 twins enrolled in the study. If 61 mothers had twins, wouldn’t the total number of twins be 122? Please clarify.
It is not double because there are 3 triplets and 1 quadruplets.
2.5 Ethics and Data Collection – Please provide ethics approval number. Again, “Figure_questionnaire” not visible to reviewer nor correctly labelled. I would suggest if including, please add this to supplementary materials.
The sentence relating to the ethical data has been deleted.
The reference "figure 2" has been inserted and additional material has been added.
Authors say all data collected was recorded in a dedicated database – is this database public? Does it have a name?
The database is internal and this clarification has been made
Not necessary to include information about meetings.
It has been eliminated
Results –
Could remove sentence “The results are summarized in tables 1-5.”. Instead, refer to the tables throughout results text. To notice that table 5 is not visible to reviewer.
The sentence has been removed
3.1. Population - Please separate results description by table. For example, include table that showcases the data that “68 (54.4%) of twin babies are first children…”.
3.2. Breastfeeding and use of MM – please include data tables here as well, if applicable. If not data tables available for these results, please create them.
The table was created
Both sections 3.1. and 3.2. could be more clearly written to showcase results. Lack of flow in writing.
3.3. – For quotations, please refer to instructions here to improve flow: https://apastyle.apa.org/style-grammar-guidelines/citations/quoting-participants
We referenced and made some changes.
Sentence, “In their story, about 30%...bottle” – how was this analyzed? Is it derived from results table. Could re-write to ”less than half of the mothers”, and remove “in their story”.
This is the modified sentence
Less than half of the mothers, once at home, continued to pump the milk and give it with a bottle.
Tables with results - Results tables lack sufficient descriptive titles that would allow reader to view them independently of text. Such tables must benefit of changing into pies or graphs?
Please change to 1.28%. In English we use periods (.) instead of commas (,), for decimal points.
Change made
In the table legend “Not Disponible Date” needs to be changed to N/A (which means not applicable, not available or no answer).
Discussion - This section is excessive in length. Many of the authors results interpretations could be moved to results section and combined with current results, including their respective headings. It would improve readability of the manuscript and results.
We have reduced the length of the discussion
We have created an ad hoc paragraph in the results
Conclusion - The first sentence should be broken up to smaller sentences and grammatically edited for readability. Also instead of a generalized affirmation “The incidence of twin births has doubled in 6 years…” should be replaced by “ The incidence in our region of…”.
The first sentence has been improved.
The replacement has been performed.
“The neonatal factors that impacted breastfeeding rates were birth weight and gestational age with the best performance observed for the lower weight and lower gestational age categories of newborns.”- the underlined observation requires a deeper introspection by the authors on other social/cultural/psychological and/or anthropological reasons why this happens, and cannot be attributed solely to a standard protocol (see examples of micro-premies in Brazil, where the NICUs have no incubators available and the babies thrive and survive on skin-to-skin care only).
“The main reasons for interrupting breastfeeding are the insufficient milk production and stress and difficulties in managing the twins.” – do the authors think of possible relationship about a ‘certain’ type of stress in relation to the type of delivery (C/S versus vaginal?), the timing or persistence of the stress? The timing and or the persistence of the support?, generally all these affirmations leave the audience wondering about what possibly could be done to change the outcome, this ‘food for thoughts’ should be delineated by the authors.
More than 90% of the deliveries were through CS so it was not possible to assess the level of stress related to this variable.
We have added this sentence:
Support implementation programs by health professionals in the first months after delivery for stress containment are desirable.
Acknowledgments - I suggest the following: “This study is dedicated to a special mother who donated her milk to the premature babies in our hospital and to those mothers contributing in donating their breastmilk to the Milk Bank of the “Bambino Gesù” Hospital in Rome. We thank all individuals who advocate and work to ensure that newborns and infants can have the right to a health opportunity represented by accessibility to human milk.”
The proposed change has been made.
This study is dedicated to a special mother who donated her milk to the premature babies in our hospital and to those mothers contributing in donating their breastmilk to the Milk Bank of the “Bambino Gesù” Hospital in Rome. We thank all individuals who advocate and work to ensure that newborns and infants can have the right to a health opportunity represented by accessibility to human milk.
Overall
Major English writing revisions are still needed, though authors have greatly improved upon their initial manuscript.
Overabundance of minor grammatical errors are present throughout the manuscript.
Few examples: “formulated milk” should be replaced by “formula milk”;
“Fertilization pregnancy” should be replaced by: “assisted reproductive technology (ART)” and so on.
“In the qualitative part of the study, the narration of the mothers of the twins reveals…” should be replaced by “ Qualitative analysis of maternal narrative revealed…”;
The change indicated has been made
“This triggered regrets and guilt. This led to the abandonment of breastfeeding in the first 2-3 months.” Should be made a continuous phrase like: “ This context triggered regrets and guilt, and led to the abandonment of breastfeeding in the first 2-3 months”.
The sentence “The limit of this study” is too optimistic, I would change to “One of the limits of this study…”. Thanks.
Please MUST change: “the pleasure of attaching them to the breast..”, we do not attach babies on the breast we make them LATCH to the breast! THANK YOU!
These changes have been added
The annexes are:
- Tables: 1 word document with tables and histograms
- Two images: Fig.1 which corresponds to the algorithm and Fig. 2 which corresponds to the questionnaire
- Tables:
- education,
- birth weight 1,2,
- gestational Age 1,2,
- MAP,
- maternal age 1,2,
- nationality,
- parity,
- profession,
- type of birth
Reviewer 2 Report
Review of Feeding the twins with breast milk. Quantitative and qualitative study in a NICU
in Southern Italy with the milk bank.
This version 2 of the manuscript is now a bit more useful, but it is still not assembled or structured correctly up to international scientific reporting standards. Nevertheless the authors have indeed effected many good corrections, which is a good sign of progress.
To further strengthen the manuscript, the authors can consider the following critiques:
Please replace “medicine student” with author affiliation.
Title – Would be better to revise title to make one sentence only. Authors could use the colon “:” to connect current sentences.
Abstract:
“The WHO and the others” seems too ambiguous. Perhaps something like “the current scientific consensus is … ”.
There is no need to have “Study:” in the abstract. Rather include a “Methods:” subsection title.
Please consolidate grammar throughout abstract, e.g., after the colon, ‘:’ this abstract text could have first letter capitalized throughout (as it is not a title of the article itself.
Introduction:
Page 2 – “The latter data…NICU” – no reference for sentence. Please include one.
“In our experience the myth…” – please rephrase sentence to refer to interpretation of your data only. Science does not deal with myths or excorcism, it deals with data and testing hypotheses.
Heading “The aim of the study” is unnecessary.
Methods:
2.1. Study Design – “thanks to the data” – could be rephrased. Scientists do not thank data as they are inanimate objects.
2.3. Study Participants – Please revise paragraph to reduce the minor grammatical errors, e.g. instead of “to preliminary information” it should be “for”.
What is “Figure_algorithm”? If Figure is available (not visible to the reviewer), please title it Figure 1.
Authors state that 61 mothers of twins completed the questionnaire, indicating that the total was 127 twins enrolled in the study. If 61 mothers had twins, wouldn’t the total number of twins be 122? Current text is confusing if not misleading.
2.5 Ethics and Data Collection – Please provide ethics approval number. Again, “Figure_questionnaire” not visible to reviewer nor correctly labelled. I would suggest if including, please add this to supplementary materials.
Authors say all data collected were recorded in a dedicated database – is this database public? Does it have a name?
Not necessary to include information about meetings.
Results:
Could remove sentence “The results are summarized in tables 1-5.”. Instead, refer to the tables throughout results text.
3.1. Population - Please separate results description by table. For example, include table that showcases the data that “68 (54.4%) of twin babies are first children…”. Use ‘were first…’
3.2. Breastfeeding and use of MM – please include data tables here as well, if applicable. If you do not yet have data tables available for these results, please create them now for R2.
Both sections 3.1. and 3.2. could be more clearly written to showcase results. There is still an awkward lack of flow in the writing.
3.3. – For quotations, please refer to instructions here to improve flow: https://apastyle.apa.org/style-grammar-guidelines/citations/quoting-participants
Sentence, “In their story, about 30%...bottle” – how was this analyzed? Is it derived from results table? You could try to re-write to ”less than half of the mothers”, and remove “in their story”.
Results tables lack sufficient descriptive titles that would allow reader to view them independently of text. This is a standard convention for all scientific reports.
Discussion:
This section is excessive in length. Many of the authors’ results and interpretations could be moved to Results section and combined with current results, including their respective headings. It would improve readability of the overall manuscript and the upshot of your results.
Conclusion section:
The first sentence should be broken up to smaller sentences and grammatically edited for readability.
Overall:
English writing revisions are still needed, although the authors have greatly improved upon their initial manuscript. Minor grammatical errors are still present throughout the manuscript, you need to improve to bring it all up to international scientific reporting conventions..
Main concern is readability and interpretability of results and discussion. The tables are not comprehensive enough, including their titles and descriptions. Many results interpretations should be moved from discussion to results section.
[END of reviewer’s critique]

Author Response
Review of Feeding the twins with breast milk. Quantitative and qualitative study in a NICU
in Southern Italy with the milk bank.
This version 2 of the manuscript is now a bit more useful, but it is still not assembled or structured correctly up to international scientific reporting standards. Nevertheless the authors have indeed effected many good corrections, which is a good sign of progress.
To further strengthen the manuscript, the authors can consider the following critiques:
Please replace “medicine student” with author affiliation.
Title – Would be better to revise title to make one sentence only. Authors could use the colon “:” to connect current sentences.
These changes have been made
Abstract:
“The WHO and the others” seems too ambiguous. Perhaps something like “the current scientific consensus is … ”.
The change has been added
There is no need to have “Study:” in the abstract. Rather include a “Methods:” subsection title.
The title has been eliminated
Please consolidate grammar throughout abstract, e.g., after the colon, ‘:’ this abstract text could have first letter capitalized throughout (as it is not a title of the article itself.
The adjustments indicated have been made
Introduction:
Page 2 – “The latter data…NICU” – no reference for sentence. Please include one.
The sentence has been deleted
“In our experience the myth…” – please rephrase sentence to refer to interpretation of your data only. Science does not deal with myths or excorcism, it deals with data and testing hypotheses.
The word myth has been replaced with “belief”
Heading “The aim of the study” is unnecessary.
The title of the paragraph has been deleted
Methods:
2.1. Study Design – “thanks to the data” – could be rephrased. Scientists do not thank data as they are inanimate objects.
Twins birth data (delivery time, parity, birth weight) are obtained using the data collection systems present at NICU (Neocare –NICU, SISWEB-regional informatic system).
2.3. Study Participants – Please revise paragraph to reduce the minor grammatical errors, e.g. instead of “to preliminary information” it should be “for”.
Changes have been made
What is “Figure_algorithm”? If Figure is available (not visible to the reviewer), please title it Figure 1.
The figure was titled
Authors state that 61 mothers of twins completed the questionnaire, indicating that the total was 127 twins enrolled in the study. If 61 mothers had twins, wouldn’t the total number of twins be 122? Current text is confusing if not misleading.
It is not double because there are 3 triplets and 1 quadruplets
2.5 Ethics and Data Collection – Please provide ethics approval number. Again, “Figure_questionnaire” not visible to reviewer nor correctly labelled. I would suggest if including, please add this to supplementary materials.
The sentence relating to the ethical data has been deleted.
The reference "figure 2" has been inserted and additional material has been added.
Authors say all data collected were recorded in a dedicated database – is this database public? Does it have a name?
Not necessary to include information about meetings.
The database is internal and this clarification has been made
Results:
Could remove sentence “The results are summarized in tables 1-5.”. Instead, refer to the tables throughout results text.
The sentence has been removed
3.1. Population - Please separate results description by table. For example, include table that showcases the data that “68 (54.4%) of twin babies are first children…”. Use ‘were first…’
The separation has been inserted and the dedicated table has been added
3.2. Breastfeeding and use of MM – please include data tables here as well, if applicable. If you do not yet have data tables available for these results, please create them now for R2.
The table was created
Both sections 3.1. and 3.2. could be more clearly written to showcase results. There is still an awkward lack of flow in the writing.
An attempt was made to make the description of the results more fluid. We have added tables
3.3. – For quotations, please refer to instructions here to improve flow: https://apastyle.apa.org/style-grammar-guidelines/citations/quoting-participants
We referenced and made some changes.
Sentence, “In their story, about 30%...bottle” – how was this analyzed? Is it derived from results table? You could try to re-write to ”less than half of the mothers”, and remove “in their story”.
This is the modified sentence
Less than half of the mothers, once at home continued to pump the milk and give it with a bottle.
Results tables lack sufficient descriptive titles that would allow reader to view them independently of text. This is a standard convention for all scientific reports.
We have improved the table titles
Discussion:
This section is excessive in length. Many of the authors’ results and interpretations could be moved to Results section and combined with current results, including their respective headings. It would improve readability of the overall manuscript and the upshot of your results.
We moved the results to another paragraph that was created exnovo
Conclusion section:
The first sentence should be broken up to smaller sentences and grammatically edited for readability.
The first sentence has been improved.
The replacement has been performed.
Overall:
English writing revisions are still needed, although the authors have greatly improved upon their initial manuscript. Minor grammatical errors are still present throughout the manuscript, you need to improve to bring it all up to international scientific reporting conventions..
We have improved the English and the text
Main concern is readability and interpretability of results and discussion. The tables are not comprehensive enough, including their titles and descriptions. Many results interpretations should be moved from discussion to results section.
Changes have been made to the text and tables following the directions
The annexes are:
- Tables: 1 word document with tables and histograms
- Two images: Fig.1 which corresponds to the algorithm and Fig. 2 which corresponds to the questionnaire
- Tables:
- education,
- birth weight 1,2,
- gestational Age 1,2,
- MAP,
- maternal age 1,2,
- nationality,
- parity,
- profession,
- type of birth